# The ratio of adaptive to innate immune cells differs between genders and associates with improved prognosis and response to immunotherapy

Johanne Ahrenfeldt[1,2,3]*, Ditte S. Christensen[1,2,4], Andreas B. Østergaard[2], Judit Kisistók[1,2,3], Mateo Sokač[1,2,3], Nicolai J. Birkbak[1,2,3]*

1 Department of Molecular Medicine, Aarhus University Hospital, Aarhus, Denmark, 2 Department of Clinical Medicine, Aarhus University, Aarhus, Denmark, 3 Bioinformatics Research Centre, Aarhus University, Aarhus, Denmark, 4 Department of Clinical Oncology, Aarhus University Hospital, Aarhus, Denmark

* ja@clin.au.dk (JA); nbirkbak@clin.au.dk (NJB)

## Abstract

Immunotherapy has revolutionised cancer treatment. However, not all cancer patients benefit, and current stratification strategies based primarily on PD1 status and mutation burden are far from perfect. We hypothesised that high activation of an innate response relative to the adaptive response may prevent proper tumour neoantigen identification and decrease the specific anticancer response, both in the presence and absence of immunotherapy. To investigate this, we obtained transcriptomic data from three large publicly available cancer datasets, the Cancer Genome Atlas (TCGA), the Hartwig Medical Foundation (HMF), and a recently published cohort of metastatic bladder cancer patients treated with immunotherapy. To analyse immune infiltration into bulk tumours, we developed an RNAseq-based model based on previously published definitions to estimate the overall level of infiltrating innate and adaptive immune cells from bulk tumour RNAseq data. From these, the adaptive-to-innate immune ratio (A/I ratio) was defined. A meta-analysis of 32 cancer types from TCGA overall showed improved overall survival in patients with an A/I ratio above median (Hazard ratio (HR) females 0.73, HR males 0.86, P < 0.05). Of particular interest, we found that the association was different for males and females for eight cancer types, demonstrating a gender bias in the relative balance of the infiltration of innate and adaptive immune cells. For patients with metastatic disease, we found that responders to immunotherapy had a significantly higher A/I ratio than non-responders in HMF (P = 0.036) and a significantly higher ratio in complete responders in a separate metastatic bladder cancer dataset (P = 0.022). Overall, the adaptive-to-innate immune ratio seems to define separate states of immune activation, likely linked to fundamental immunological reactions to cancer. This ratio was associated with improved prognosis and improved response to immunotherapy, demonstrating potential relevance to patient stratification. Furthermore, by demonstrating a significant difference between males and females that associates with response, we highlight an important gender bias which likely has direct clinical relevance.

**Data Availability Statement:** Data are available from the Hartwig Medical Foundation and an application to access the data can be sent here https://www.hartwigmedicalfoundation.nl/en/data/data-acces-request/ for researchers who meet the criteria for access to confidential data. The TCGA data can be accessed here: https://www.cancer.gov/about-nci/organization/ccg/research/structural-genomics/tcga The dataset we used is the TCGA project.

**Funding:** NJB is a fellow of the Lundbeck Foundation (R272-2017-4040), and acknowledges funding from Aarhus University Research Foundation (AUFF-E-2018-7-14), and the Novo Nordisk Foundation (NNF21OC0071483). The funders had no role in study design, data collection and analysis, decision to publish, or preparation of the manuscript.

**Competing interests:** The authors have declared that no competing interests exist.

## Introduction

Metastatic disease is the leading cause of cancer-related death. Metastasis is the process where cancer cells from a primary site colonise to distant organs [1], and is usually considered the terminal step in the evolution of lethal cancer. Primary cancer can in most cases be surgically removed, when it is categorised as local disease. In these cases, the patients often have a good prognosis. It has been hypothesised that the ability to metastasise is not inherent to primary cancers, but must be acquired during cancer evolution [2]. We lack understanding of how and when during cancer evolution the primary tumour achieves metastatic potential. In order to improve the survival of cancer patients this knowledge is of critical importance. Studies during the last decade have suggested that the non-cancer cells within the tumour microenvironment play an important role in the development of metastatic disease and response to immunotherapy [3–5].

The immune system has a critical role in cancer. It utilises T-cells to clear cancer cells harbouring novel cancer neoantigens, but it also provides some of the necessary mechanisms for developing cancer, e.g., by hijacking the inflammatory system to promote tumour growth by secretions of pro-survival, pro-migration and anti-detection factors [2].

It is now well-known that the activation of T-cells and the expression of immunoinhibitory checkpoints such as CTL-antigen (CTLA-4) and Programmed Cell Protein 1 (PD-1) play an important role in the anti-tumour response of the immune system. These molecules are expressed on T-cells, but when they bind to their ligands on antigen-presenting cells the anti-tumour response is suppressed. Checkpoint inhibitor (CPI) immunotherapy specifically Inhibits this interaction.

The immune system's ability to combat cancer has been utilised in immunotherapy, which has recently revolutionised the treatment of metastatic cancer. E. g. in metastatic melanoma as many as half of all patients treated by CPI therapy are long-term survivors [6], where until recently, patients rarely lived past the half-year milestone. Despite the improved survival, many patients still die from their disease, and although CPI therapies are applied across cancer types, the response rates are significantly lower outside of melanoma [7]. A great amount of research is being performed to understand the dynamics of the immune response, to identify biomarkers of immunotherapy response, and to characterise which patients are sensitive to the treatment. Inflammation is the body's response to tissue damage, and the inflammatory environment is characterised by the presence of host leukocytes and supporting stroma around the tumour. Solid tumours grow into normal tissue and recruit diverse mesenchymal and inflammatory cells. These are found both inside the primary tumour and in the tumour vicinity, and together with the neoplastic cells they form the tumour-associated stroma referred to as the tumour microenvironment (TME) [8]. Tumour associated macrophages (TAM) are one of the most abundant cell types in the TME [9]. When activated, tumour-associated macrophages can kill cancer cells and thereby play an important role in the defence against cancer cells. However, they can also stimulate tumour cell proliferation, promote angiogenesis and favour metastatic dissemination [9]. Indeed, it has previously been described how infiltration of TAM and regulatory T-cells is associated with poor prognosis, whereas infiltration of CD8+ T-cells is associated with improved outcome [10].

The immune system can roughly be divided into two major branches, the innate and the adaptive. The innate immune system is our first line of defence, but it is non-specific and its primary role is to initiate inflammation when recognizing foreign pathogens, and to use phagocytosis to engulf foreign molecules and cells, and then present antigens from these to the cells of the adaptive immune system that can activate a specific immune response [11]. The adaptive immune system contains cells that undergo recombination to create unique receptors which bind to foreign peptides or peptides not usually presented by normal, healthy cells [12].

Activation of the innate and adaptive Immune system varies between genders. Females have been shown to induce a higher production of IFN-α after stimulation with toll-like receptor 7 from plasmacytoid dendritic cells, which leads to a stronger secondary activation of CD8 + T cells. Female hormones cause delay of neutrophil apoptosis. Conversely, monocytes from males produce higher levels of pro-inflammatory cytokines after LPS stimulation. Macrophage polarisation differs between males and females and during placental development the decidual NK cells are involved in tissue remodelling [13]. Combined, these differences are associated with a higher male mortality rate from infectious diseases, while in females a relatively higher incidence of autoimmune diseases and a better response to vaccines are observed [13, 14]. There is also a difference in the cancer incidence between genders, a global study found that males have a significantly higher incidence of cancer at 32 of 35 studied sites of the body compared to females [15].

Therefore, to investigate the prognostic impact of tumour infiltration by adaptive and innate immune cells and how this may vary by gender, we divided immune cells into adaptive and innate categories, and calculated the ratio of adaptive-to-innate immune infiltration for each patient. For this study, Dendritic cells, Macrophages, Mast cells, Neutrophils, Natural killer cells and Natural killer CD56dim cells were all analysed as part of the innate immune system. Likewise, CD8 T-cells, B-cells, CD45, Cytotoxic cells, T-cells, Th1-cells and T-regulatory cells were all analysed as part of the adaptive immune system [11, 12].

This allowed us to investigate outcome and therapy response in relation to a simple ratio obtained from immune cells in the tumour microenvironment. Based on this analysis, we found that a higher level of infiltration of adaptive relative to innate immune cells associated with improved outcome and improved response to immunotherapy.

## Methods

### Data acquisition and preprocessing

Clinical information from 11,162 sequenced tumour samples was acquired from The Cancer Genome Atlas [16]. Gene expression data which had been uniformly normalised for all samples was acquired from the UCSC Xena database [17]. Cancers involving immune cells and tissues related to the immune system were omitted from the analysis (LAML, DLBC and THYM). Cancer type abbreviations are found in Table 1. Information regarding MSI status [18] in colon cancer was used to split the COAD patients into COAD MSI, COAD MSS and COAD, the latter for the patients where the information was not available.

Gene expression data and sample information from 1406 cancer cell line samples from the Cancer Cell Line Encyclopedia was acquired from DepMap [19].

Clinical information and normalised gene expression data summarised to genes using transcript per million (TPM) from 1759 metastatic tumour samples was acquired from the Hartwig Medical Foundation (HMF) [20]. Clinical information and raw RNAseq data from 348 metastatic tumour samples from bladder cancer patients treated with immunotherapy was acquired from the European Genome-Phenome Archive EGAS00001002556 [21]. The raw RNAseq data was aligned against the human genome hg38 using STAR [22] version 2.7.2 and processed to generate transcript per million (TPM) expression values using Kallisto [23] version 0.46.2. For the two cohorts of Checkpoint Inhibitor treated patients, only patients with complete response (CR), partial response (PR) and progressive disease (PD) were used in the analysis of response. Patients with stable disease (SD) were not included in the analysis, as the interpretation of SD is not clearly defined as good or poor outcome. Indeed, it can be both a sign that the therapy works and contains tumour growth, or it can be a sign that the therapy has no effect but the tumour size remains unchanged due to stagnated growth.

**Table 1. Cancer type abbreviations from the Cancer Genome Atlas.**

| Abbreviation | Cancer type |
|---|---|
| LAML | Acute Myeloid Leukemia |
| ACC | Adrenocortical carcinoma |
| BLCA | Bladder Urothelial Carcinoma |
| LGG | Brain Lower Grade Glioma |
| BRCA | Breast invasive carcinoma |
| CESC | Cervical squamous cell carcinoma and endocervical adenocarcinoma |
| CHOL | Cholangiocarcinoma |
| COAD | Colon adenocarcinoma Microsatellite Unknown |
| COAD MSI | Colon adenocarcinoma Microsatellite Instability |
| COAD MSS | Colon adenocarcinoma Microsatellite Stable |
| ESCA | Esophageal carcinoma |
| GBM | Glioblastoma multiforme |
| HNSC | Head and Neck squamous cell carcinoma |
| KICH | Kidney Chromophobe |
| KIRC | Kidney renal clear cell carcinoma |
| KIRP | Kidney renal papillary cell carcinoma |
| LIHC | Liver hepatocellular carcinoma |
| LUAD | Lung adenocarcinoma |
| LUSC | Lung squamous cell carcinoma |
| DLBC | Lymphoid Neoplasm Diffuse Large B-cell Lymphoma |
| MESO | Mesothelioma |
| MISC | Miscellaneous |
| OV | Ovarian serous cystadenocarcinoma |
| PAAD | Pancreatic adenocarcinoma |
| PCPG | Pheochromocytoma and Paraganglioma |
| PRAD | Prostate adenocarcinoma |
| READ | Rectum adenocarcinoma |
| SARC | Sarcoma |
| SKCM | Skin Cutaneous Melanoma |
| STAD | Stomach adenocarcinoma |
| TGCT | Testicular Germ Cell Tumors |
| THCA | Thyroid carcinoma |
| THYM | Thymoma |
| UCS | Uterine Carcinosarcoma |
| UCEC | Uterine Corpus Endometrial Carcinoma |
| UVM | Uveal Melanoma |

## Tumour infiltrating leukocytes

Tumour immune cell decomposition was performed using the score defined by Danaher and colleagues [24] based on whole tumour RNAseq data, implemented as described [25]. We used a defined list of genes from Danaher [24] to define the expression of immune cell types, and the mean of the cell types described in the paper was then used as the total TIL score [25]. The immune cell scores and the following TIL score was calculated based on the genes found in S1 Table.

## Adaptive innate ratio

To calculate the A/I ratio, the immune cells were divided into adaptive immune cells and innate immune cells, based on which overall compartment they belong to in the immune

system. Adaptive immune cell types were defined as: CD8 T-cells, B-cells, CD45, Cytotoxic cells, T-cells, Th1-cells and T-regulatory cells. Innate immune cell types were defined as: Dendritic cells, Macrophages, Mast cells, Neutrophils, Natural killer cells and Natural killer CD56dim cells. A linear scaling of the expression values for each cell type was performed as follows: first the values were reverse log-transformed, and then the values within each cell type were linearly scaled to values between 0 and 1, with this equation:

$$\text{scaled\_celltype}_n = \frac{\text{celltype}_n - \text{celltype}_{min}}{\text{celltype}_{max} - \text{celltype}_{min}} \qquad \text{Eq 1}$$

and then a score for each group (adaptive and innate) was calculated per sample as the mean scaled value for the cell types within the group, whereafter the A/I ratio was determined by dividing the adaptive with the innate score.

### Ranked expression of immune genes

For each sample all genes were assigned a rank based on their expression level. The lowest expression got the rank 1, the second lowest 2 etc. If more genes had the same value, the rank was averaged. The immune genes were extracted, and for each cancer type, a mean rank for each gene was calculated.

### Statistical analyses

All data analysis was performed in R version 3.6.3 [26], using tidyverse [27], survminer [28], survival [29], scales [30], ggbeeswarm [31] and Publish [32]. Survival analyses were performed by Cox proportional hazard regression [33] and Kaplan-Meier curves. Significance testing of differences between groups was performed by Wilcoxon test, unless otherwise mentioned. Fisher's exact test was used to determine if the number of responders was higher in a subset of the data. All p-values are two-sided.

## Results

### Composition of the tumour immune cell microenvironment associates with outcome

To investigate the relevance of the immune cell composition in the tumour microenvironment relative to disease progression, we performed immune cell decomposition of 8,024 tumours across 25 cancer types from the Cancer Genome Atlas (TCGA). We then fitted a multivariable Cox proportional hazard model to the progression free interval, including all immune cell types and gender, age and tumour stage as covariates (Fig 1A omitting age, stage and gender from the visualisation. Full results with all covariates listed in S2 Table). Of 14 immune cell types, 8 showed a significant association with outcome, four with improved survival, four with worse survival. Overall, we observed that adaptive immune cells associated with a lower risk of relapse (CD8 T-cells, CD45, T-cells, Th1 cells, Treg), while innate immune cells were associated with a higher risk of relapse (Dendritic cells, Macrophages, Natural killer cells) (Fig 1B). When we performed the same analysis including cancer types as covariates, the same overall pattern was observed with regard to the direction of association of the individual immune cell types, although unsurprisingly cancer type was by far the most significant covariates relative to outcome reflecting established cancer-type specific prognosis (S1A Fig). To further evaluate how the two compartments of the immune system associate with patient outcome in opposite directions, we stratified the cell types based on which of the two major immune components they belonged to and calculated a value for each component, adaptive and innate. We then

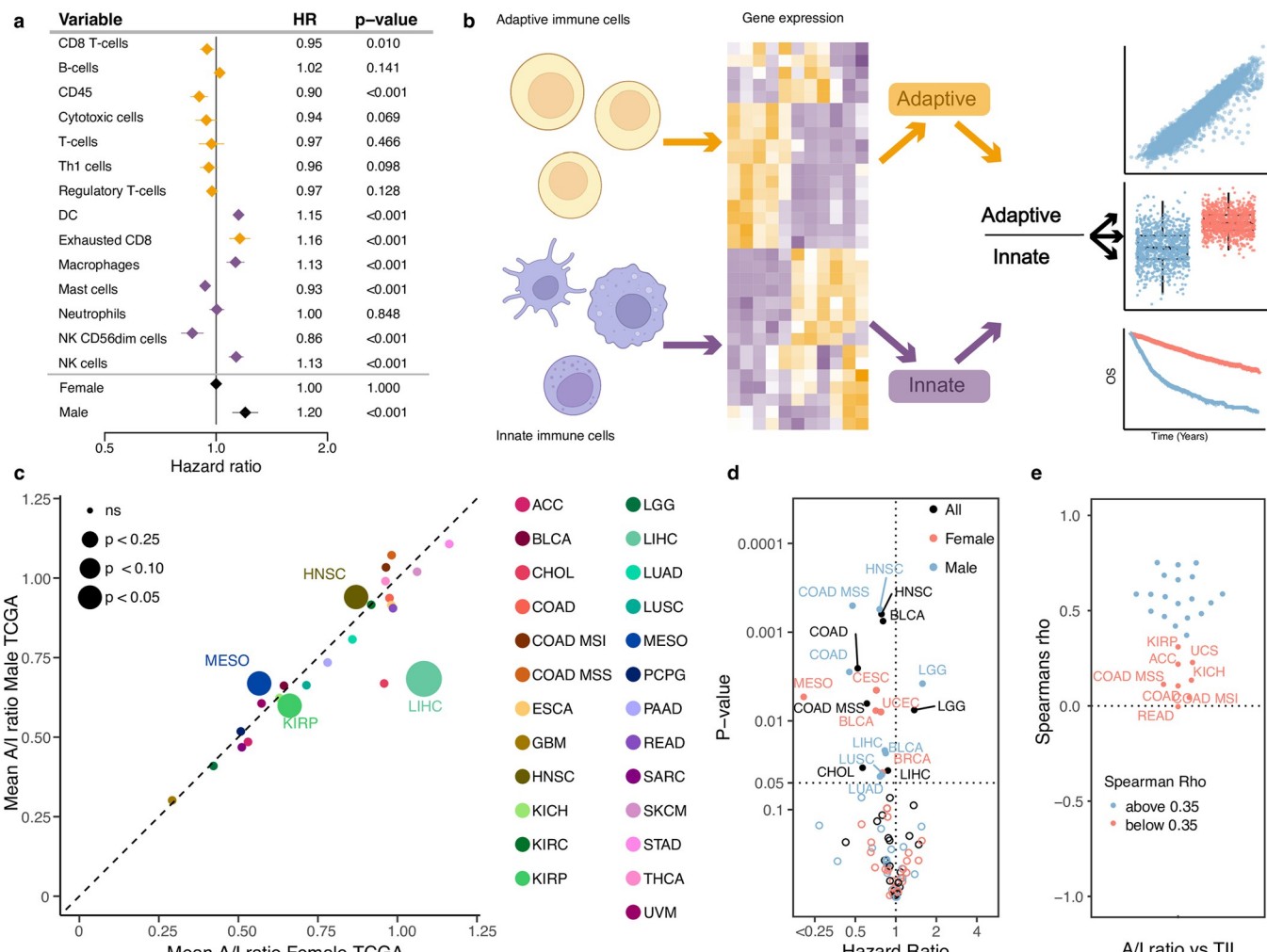

**Fig 1. Calculating the adaptive/immune ratio.** a) A forest plot showing the hazard ratio for progression of cancer for the expression of each of the cell types in the TIL calculation. Yellow marks adaptive immune cells, purple marks the innate immune cells. b) A schematic showing how we calculate the A/I ratio. c) The mean A/I ratio for 29 cancer types in the TCGA cohort. The female mean ratio on the x-axis and male ratio is in the y axis. The size of the dot represents the p-value for the comparison of the mean. d) Univariate cox proportional hazard regression for the A/I ratio against progression free interval (PFI) across cancer types and genders for TCGA. e) Correlation of the A/I ratio and the TIL score across all cancer types in TCGA.

performed a multivariate model including the adaptive and innate values together with age, stage and gender. This was done separately pan-cancer and with cancer type as covariates. We found that a high adaptive component was significantly associated with improved outcome (pan-cancer: HR = 0.016, P = $9.20 \times 10^{-7}$, cancer informed: HR = 0.071, P = 0.0014,) whereas a high innate component was associated with poor outcome, although only significant in pan-cancer (pan-cancer: HR = 80.9, P = $1.94 \times 10^{-6}$, cancer informed: HR = 1.042, P = 0.57), potentially indicating that the innate component may be of less relevance to outcome within cancer types. To further investigate the observation that high expression of adaptive immune genes and low expression of innate immune genes is associated with improved outcome, we calculated the ratio of Adaptive to Innate immune cells (A/I ratio, see methods, Fig 1B). We then compared the A/I ratio within cancer types, across the TCGA cancer cohort. While we observed a large variation in the A/I ratio, ranging from a mean of 0.25 in GBM to a median ratio of 1.0 in PRAD (S2 Fig). When we explored gender-specific differences, we observed

largely similar ratio values between males and females (Fig 1C), with a few notable exceptions. The A/I ratio particularly shows large gender-specific variation in LIHC, where the median for female patients was more than 25% greater than in male patients, and in ACC and BLCA where the median for male patients was 15% and 14% greater than for female patients. To confirm that the expression of immune related genes was in fact originating from the TME and not from the cancer cells we explored the expression of the individual immune genes in the Cancer Cell Line Encyclopedia (CCLE) [19], a dataset of cancer cell lines (thus devoid of any infiltrating immune cells). Here we observed that the ranked expression of the immune genes was low for all cancer cell lines, except for cell lines originating from leukaemia and lymphoma, both cancers of the immune system. When we compared the ranked expression of cancer cell lines to TCGA tumour samples of matched tissue, we observed significantly higher ranks for all tumours (S3 Fig), indicating that the observed immune signal did indeed originate from infiltrating immune cells, and not from cancer cells expressing immune-related genes.

## A higher ratio of adaptive to innate immune infiltration associates with improved prognosis

To investigate the impact of the A/I ratio on prognosis across cancer types, we performed univariate Cox regression against progression-free survival. As the A/I ratio varied across and within cancer types and also by gender, the hazard ratio (HR) was calculated within each cancer type for all patients and separately by gender. We observed that a higher A/I ratio significantly associated with improved outcome in 12 cancer types (BLCA, BRCA, CESC, CHOL, COAD, COAD MSS, HNSC, LIHC, LUAD, LUSC, MESO & UCEC), supporting the known role of the adaptive immune system in combating cancer [34]. Interestingly, for COAD, COAD MSS, HNSC, LIHC, LUAD and LUSC only males showed a significant association, while for MESO, only females. Lower A/I ratio only associated with improved outcome in male LGG patients (Fig 1D, S1B Fig). Across the cohort of cancer patients, we observed that the A/I ratio showed a good correlation to the total level of TIL infiltration (mean spearman rho = 0.38, Fig 1E). To investigate if the prognostic relevance of the A/I ratio was independent of the total TIL infiltration level, we performed multivariate Cox hazard regression, where we included the A/I ratio and TIL score, along with gender, age and tumour stage. We observed that the A/I ratio was highly significant and associated with improved outcome (A/I ratio: HR = 0.77, P < $2x10^{-16}$), while the total TIL score was associated with poorer outcome (TIL HR = 1.03, P = 0.0137). When we included cancer type as a covariate in the model, both terms were significantly associated with improved outcome (AI ratio: HR = 0.92, P = 0.000988, TIL: HR = 0.94, P = 0.000792), indicating that both the specific ratio of adaptive to innate immune cells and the total amount of immune cells are independently associated with outcome.

When we performed survival analysis on the combined TCGA cohort including all patients, we found that both female and male patients with an A/I ratio above median had significantly improved overall survival relative to patients with an A/I ratio below median (Fig 2A). We performed the same analysis on the individual cancer types, and found that 7/30 cancer types (BRCA, CESC, HNSC, LICH, OV, SKCM and UCEC) showed significantly improved outcome with an A/I ratio above median, while 2/30 (LGG and UVM) showed the opposite (S4 Fig). Based on these results, we performed a meta-analysis which take all cancer types into account, on male and female patients separately. Here, we observed that an A/I ratio above median associated with improved outcome in both male and female patients, but with a stronger association in females (HR females 0.73, HR males 0.86, P < 0.05, Fig 2B and 2C). To test if the observed gender difference in survival could be explained by differences in the immune systems between male and females, we compared the outcome between males with an A/I ratio

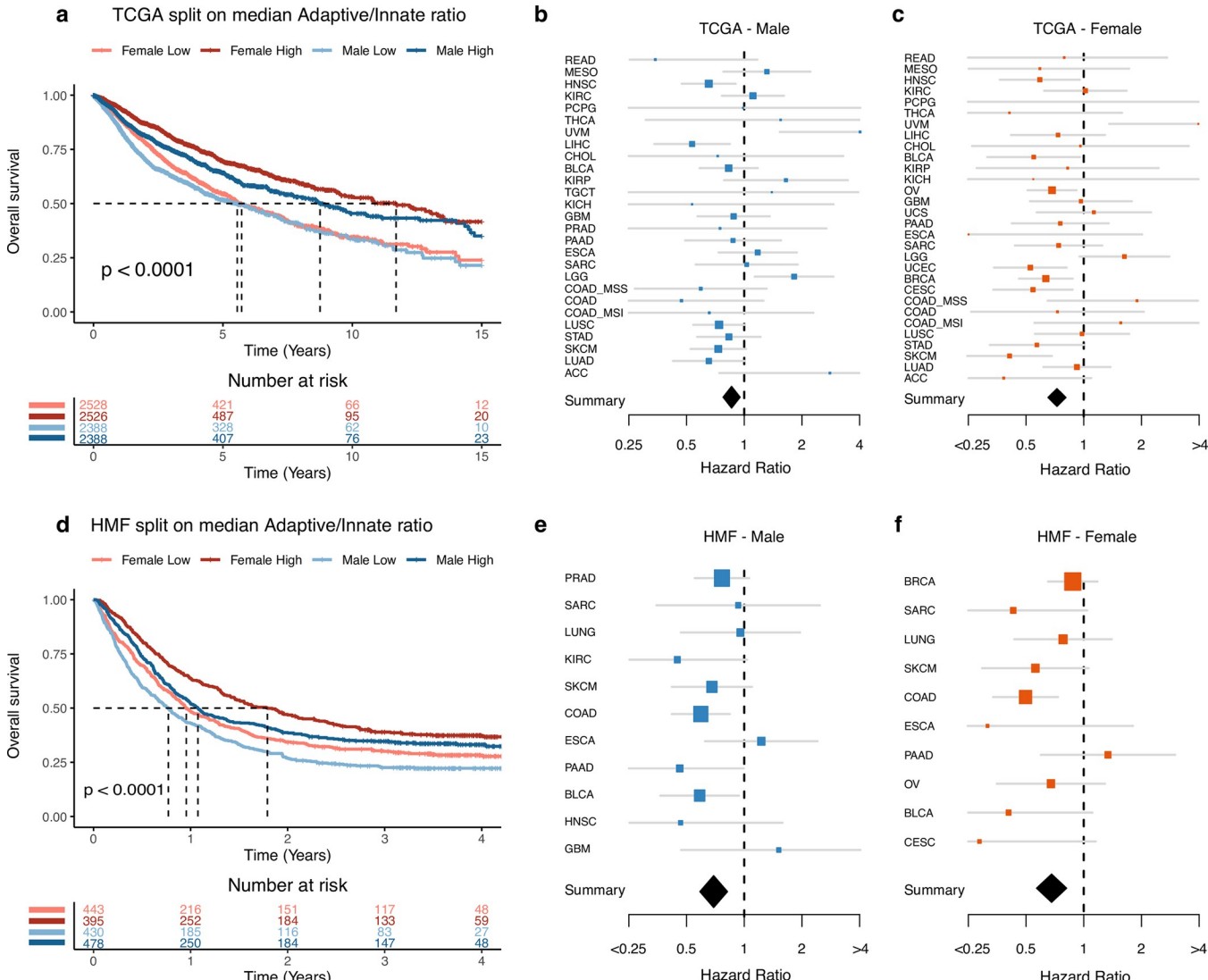

**Fig 2. Survival based on A/I ratio.** a) A Kaplan-Meier curve showing the 15-year overall survival for patients in the TCGA cohort, the patients are divided by gender and by median A/I ratio. b) A forest plot showing the univariate cox proportional hazard regression for the A/I ratio against overall survival, for the male patients in each of the TCGA cancer types. The diamond represents the hazard ratio from the meta-analysis across all cancer types. c) A forest plot showing the univariate cox proportional hazard regression for the A/I ratio against overall survival, for the female patients in each of the TCGA cancer types. The diamond represents the hazard ratio from the meta-analysis across all cancer types. d) A Kaplan-Meier curve showing the 4-year overall survival for metastatic cancer patients in the HMF cohort, the patients are divided by gender and by median A/I ratio. e) A forest plot showing the univariate cox proportional hazard regression for the A/I ratio against overall survival, for the male patients in each of the cancer types in the HMF cohort. The diamond represents the hazard ratio from the meta-analysis across all cancer types. f) A forest plot showing the univariate cox proportional hazard regression for the A/I ratio against overall survival, for the female patients in each of the cancer types in the HMF cohort. The diamond represents the hazard ratio from the meta-analysis across all cancer types.

above the female median A/I ratio, and females with an A/I ratio above median, for each cancer type. We used this data to do a survival analysis, and found that across cancer types, there were only two where survival was significantly improved in females (ESCA: P = 0.04, THCA: P = 0.011, S5 Fig). This suggests that some of the gender difference in outcome may be explained by the increased A/I ratio in females. To investigate if the differences in survival were solely based on gender, we performed two cox proportional hazard models, one analysing survival relative to gender, and one analysing survival relative to gender and the A/I ratio.

Both models had age, stage and cancer type as covariates. We then compared the performance of the models using a likelihood ratio test. Based on this analysis, we found that the model including the A/I ratio term significantly out-performed the simpler model including only gender (P = $4.45 \times 10^{-9}$).

We next asked if higher levels of innate immune infiltration may support cancer metastasis. To investigate this, we analysed the A/I ratio in 1,759 metastatic tumours from 22 different cancer types from the Hartwig Medical Foundation (HMF) dataset of metastatic cancers [20]. Initially, we performed a survival analysis on the metastatic HMF cohort and found that both male and female patients had improved overall survival if their A/I ratio was above median (Fig 2D). We performed the same analysis on the individual cancer types, and found that 2/11 cancer types (BLCA and COAD) showed significantly improved prognosis with an A/I ratio above median, while no cancer types showed the opposite (S6 Fig). When we performed a meta-analysis on male and female patients, respectively, we again found that an A/I ratio above median associated with improved outcome in both male and female patients (HR females 0.68, HR males 0.69, P < 0.05, Fig 2E and 2F).

To further investigate the observed gender differences, we specifically investigated outcome by gender. Within both cohorts we observed a gender difference in survival; female patients had significantly improved overall survival in both TCGA (P < 0.0001, S7A Fig) and HMF (P < 0.0017, S7B Fig).

Next, we compared the A/I level between cancer types in HMF and TCGA. Of the 21 comparable cancer types, 9 had a significantly higher A/I ratio in TCGA (BRCA, CESC, COAD, ESCA, KIRC, PAAD, PRAD, SKCM and UCEC) while 2 had a significantly higher A/I ratio in HMF (GBM and OV) (Fig 3). The increased level in GBM may be related to the brain's immune privileged status, preventing immune cells from infiltrating, potentially suggesting that a high A/I ratio may be one of the factors that contribute to a lower frequency of patients progressing to metastatic disease.

Taken together, we have found that in both the TCGA and the HMF cohorts, which includes tumours from patients with different cancer types treated with different protocols, patients with a higher A/I ratio showed significantly improved outcomes.

## A higher ratio of adaptive to innate immune infiltration associates with improved response to immunotherapy

The immune system plays a significant role in controlling cancer growth, and during anticancer treatment the adaptive immune system can be activated by the application of immunotherapy. To investigate how the balance of the adaptive to the innate immune response affects response to immunotherapy response, we determined the A/I ratio in 412 metastatic tumours treated with checkpoint inhibitor immunotherapy (CPI). These included 177 tumours from 6 cancer types in the HMF dataset [20], and 235 metastatic bladder cancer tumours from the Mariathasan dataset, treated with anti-PDL1 (atezolizumab) [21]. To determine if the previously observed gender difference in cancer prognosis also affects survival within the two cohorts of CPI treated patients, we performed a survival analysis on gender. For both cohorts we found no significant difference in survival of male and female patients treated by CPI (Mariathasan: P = 0.18, S7C Fig, HMF BLCA: P = 0.081, S7D Fig, HMF LUNG: P = 0.17, S7E Fig, HMF SKCM: P = 0.72, S7F Fig), indicating that drug-induced activation of the adaptive immune response may out-weigh any gender-specific differences in the immune response. To rule out any potential sampling biases, we investigated the percentages of patients from each cancertype that received immunotherapy, and found that in the HMF cohort, an equal percentage of male and female patients have received CPI for each of the cancer types (BLCA

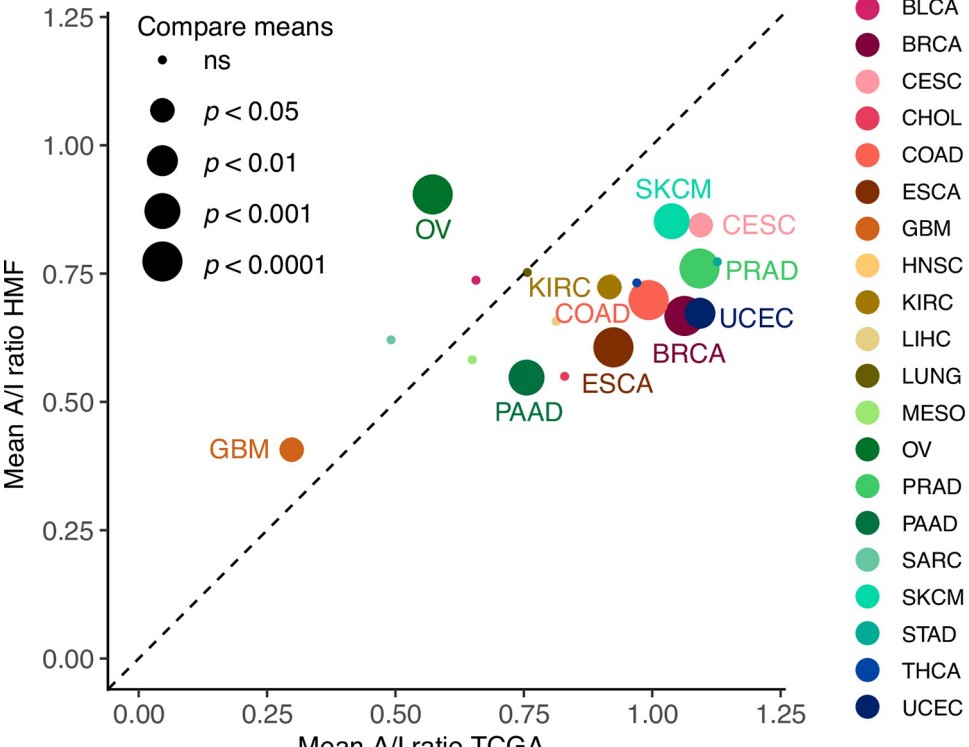

**Fig 3. Comparing the A/I ratio between primary and metastatic tumours.** The mean A/I ratio for 20 overlapping cancer types in the TCGA and HMF cohorts. The mean ratio of primary tumours from TCGA on the x-axis and mean ratio of metastatic tumours from the HMF cohort on the y-axis.

total: 78% male, CPI: 81% male. LUNG total: 45% male, CPI: 49% male. SKCM total: 60% male, CPI: 59% male), and the percentage of responders within each gender is similar within the HMF cohort. And between them there is an approximately equal distribution of each response category. The same is true for the metastatic bladder cancer cohort where we also find an equal distribution between males and females in the response groups. Taken together this indicates that these results are not affected by gender sampling bias.

To investigate if a higher A/I ratio associates with improved response to immunotherapy, we compared the A/I ratio to response across both cohorts. In the Mariathasan bladder cancer cohort, patients with complete response (CR) to immunotherapy had a higher A/I ratio than partial responders (PR) and a significantly higher A/I ratio than patients with progressive disease (PD) ($P = 0.022$, Fig 4A). When each response group is split on gender, we only observe a difference in the A/I ratio in the male CR patients ($P = 0.047$, S8A Fig). However, this group is represented by only four patients, and should therefore be interpreted with caution. No difference in response rate was observed based on gender. In the HMF dataset including multiple cancer types, both the patients with CR and PR have a significantly higher ratio than the patients with PD (CR: $P = 0.036$, PR: $P = 2.6 \times 10^{-6}$, CR+PR: $2.6 \times 10^{-6}$). There was no significant difference in the A/I ratio between CR and PR patients (Fig 4B). There were no significant differences on the A/I ratio within the response groups, when divided by gender (S8B Fig). When we restricted the HMF cohort to BLCA patients, for comparability to the Mariathasan dataset, we observed only one complete responder, and a significantly higher A/I ratio in PR patients compared to patients with PD ($P = 0.00076$). For lung cancer we observe a significantly higher ratio for CR compared to PR ($P = 0.0385$) and PD ($P = 0.0035$), and for Melanoma we observe

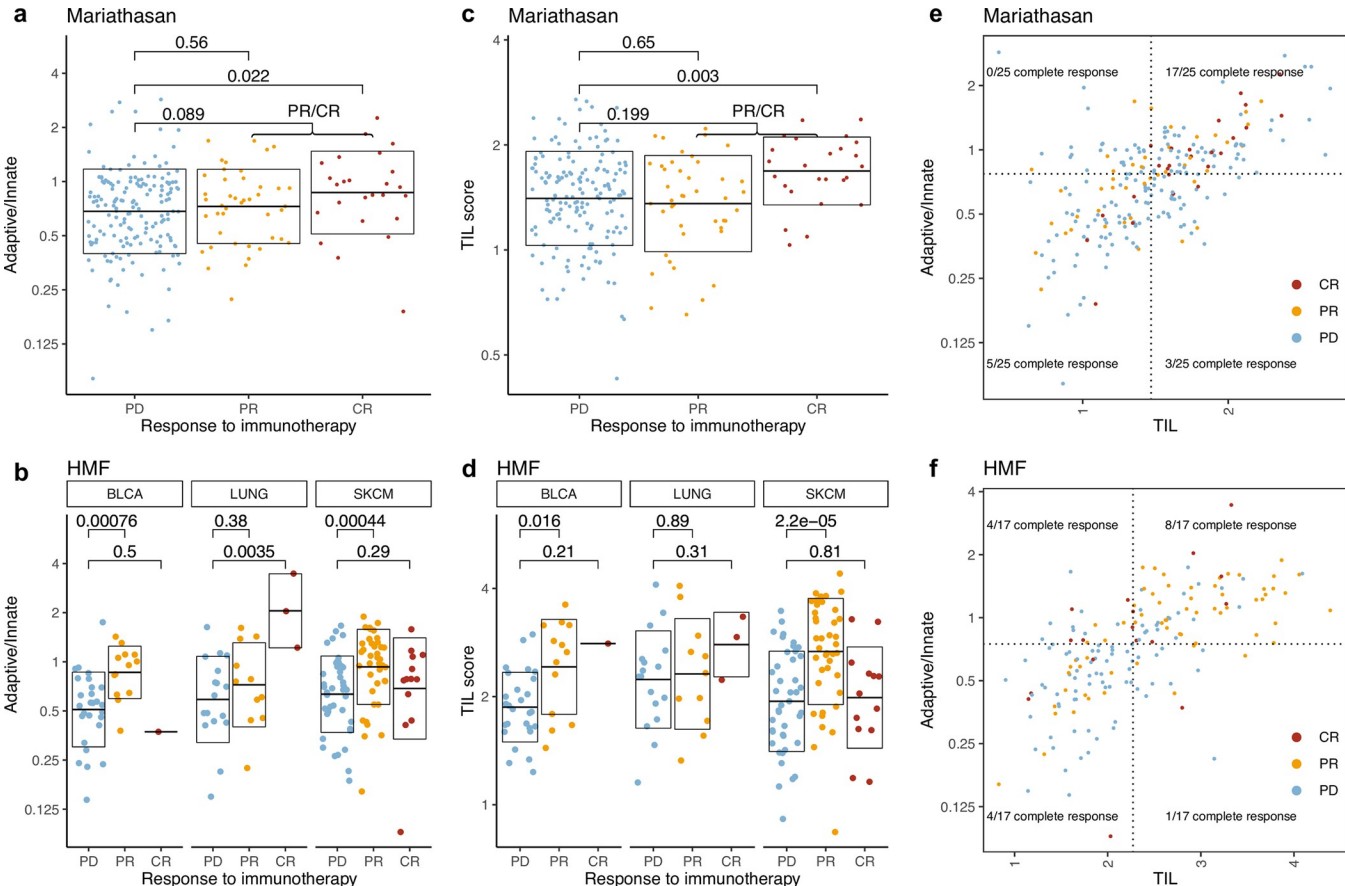

**Fig 4. Response to immunotherapy.** a) A/I ratio vs response to immunotherapy categories for the metastatic bladder cancer patients in the Mariathasan cohort. b) A/I ratio vs response to immunotherapy categories for the CPI treated metastatic cancer patients from the HMF cohort, for each of the three major cancer types. c) TIL score vs response to immunotherapy categories for the metastatic bladder cancer patients in the Mariathasan cohort. d) TIL score vs response to immunotherapy categories for the CPI treated metastatic cancer patients from the HMF cohort, for each of the three major cancer types. e) A/I ratio vs TIL score for metastatic bladder cancer patients in the Mariathasan cohort, colored by response category. 17/25 (68%) of patients with CR had both an A/I ratio and a TIL score above median (P = 0.0006). f) A/I ratio vs TIL score for the CPI treated metastatic cancer patients from the HMF cohort, colored by response category. 51/84 (61%) of all responders (CR and PR) there both have a high A/I ratio and a high TIL score (P = 1.4x10^-7).

no difference between CR and PR, but a significantly higher ratio in PR compared to PD (0.00044) (S8C Fig). We observed no significant gender differences within the cancer types and response groups following CPI therapy (alternative S8D Fig), indicating that gender differences in pre-treatment A/I ratio may be of limited relevance to response to immunotherapy.

## A/I ratio associates directly with immune infiltration and predicts response to checkpoint immunotherapy

A high A/I ratio does not in itself represent a high level of immune infiltration but may be driven by a large adaptive component of a relatively small level of infiltrating immune cells. To investigate whether the A/I ratio is indeed reflective of the immune activity that takes place within the tumour, we compared it to tumour immune phenotypes scored by immunohistochemistry, defined as inflamed, excluded and desert depending on the degree of immune infiltration [21]. We found that for both male and female patients the inflamed patients have a significantly higher A/I ratio (male: P = $1.3 \times 10^{-8}$, female: P = 0.018) and for the male patients the inflamed cohort also had a significantly higher A/I ratio than the excluded cohort

(P = 0.03) (S9A Fig). This demonstrates that the A/I ratio associates directly with higher levels of immune cell infiltration, indicating that the main component of increasing TILs is associated with adaptive immune cells. We also observed that the overall level of TIL infiltration showed a significant association with outcome (S9B Fig). On a cancer specific level, five cancer types showed improved survival with a TIL score above median (S10 Fig). Next, we investigated if the combination of a high A/I ratio and a high TIL score may show improved association with immunotherapy response. We compared the TIL score of each of the response categories and found that like the A/I score, a high TIL score was associated with response in both cohorts (Mariathasan, CR: P = 0.003, Fig 4C. HMF, PR+CR: P = $4.8 \times 10^{-6}$, Fig 4D). We defined high and low A/I ratio and high and low TIL infiltration based on the median value. In the Mariathasan cohort, 17/25 (68%) of patients with a complete response to immunotherapy had both an A/I ratio and a TIL score above median (P = 0.0006, Fig 4E). In the HMF CPI treated cohort 51/84 (61%) of patients with a response to immunotherapy (CR+PR) both have an A/I ratio and TIL score above median (P = $1.4 \times 10^{-7}$, Fig 4F). When HMF was divided by cancer type, we observed the same tendency, there was a significantly higher number of responders with an A/I ratio and TIL score above median (BLCA: P = 0.003, LUNG: P = 0.15, SKCM: P = 0.006, S9C–S9E Fig).

To further elucidate the relationship between expression of the adaptive and innate immune system in the tumour microenvironment of the CPI treated patients with progressive disease, we investigated the adaptive vs. the innate expression scores directly. Here, we observed that for both the HMF cohort and the Mariathasan bladder cancer, a very large proportion of patients with progressive disease showed higher innate relative to adaptive scores (BLCA: 96%, LUNG: 76%, SKCM: 83%, Fig 5. Mariathasan BLCA: 79%, S11 Fig). This supports the use of a ratio as an efficient tool to identify patients with relatively increased adaptive-to-innate expression, and potentially improved therapy response.

## Discussion

Our findings show that large infiltration of adaptive immune cells relative to innate immune cells leads to a better prognosis for primary cancer and to improved immunotherapy response. This is consistent with literature demonstrating an immune regulatory role of innate immune cells [35]. It has previously been reported that macrophages expressing CD163 in the tumour microenvironment can lead to a poor prognosis for cancer patients [36–38]. In our work, CD163 is one of the genes used to define macrophages, an element of the innate immune system. CD163+ macrophages have also recently been found to play a role in maintaining suppression in anti-PDL1 resistant melanoma in an experimental setting [39]. This supports our findings that 83% of patients with metastatic melanoma with a poor response to CPI have a lower A/I ratio, indicating that they have a relatively high expression of the innate immune system (Fig 5).

Consistent with literature we observe improved female outcomes relative to male [40, 41]. Interestingly this difference is no longer significant when we focus on patients treated by immunotherapy. While we recognise the CPI treated cohorts may be underpowered, we observed no trend supporting improved female-to-male response in either cohort. Within the HMF cohort, the percentage of responders within each gender is similar and an equal percentage of males and females have received immunotherapy. The same is true for the metastatic bladder cancer cohort where we also find an equal distribution between males and females in the response groups. To test if the gender difference could be explained by the difference in the immune system, we compared the outcome between men with an A/I ratio above the female median A/I ratio, and females with an A/I ratio above median, for each cancer type and found

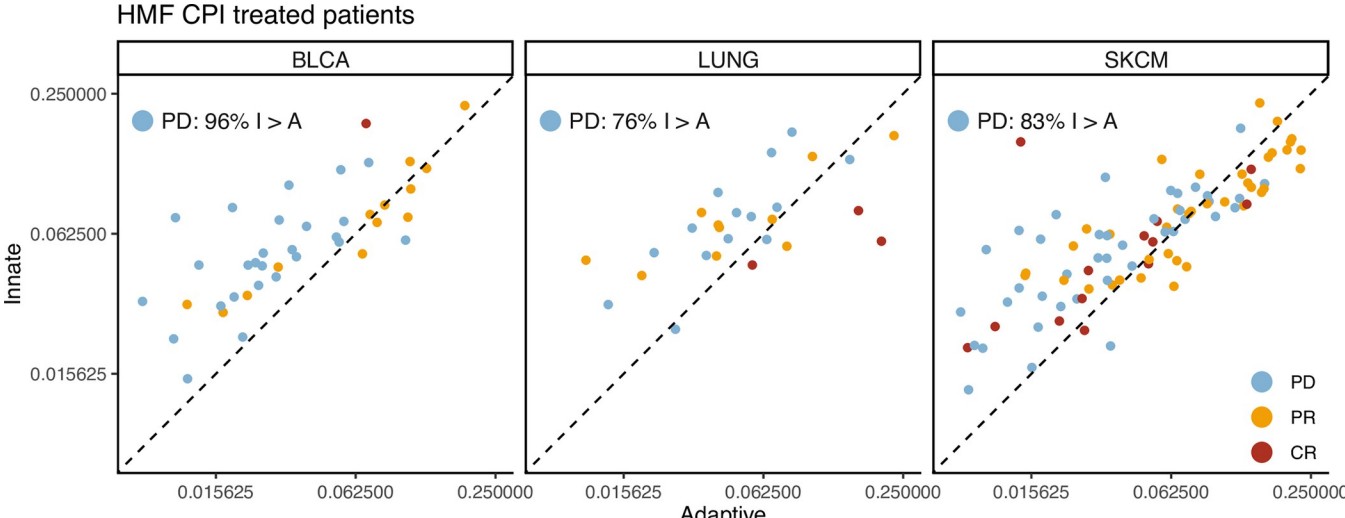

**Fig 5. Adaptive vs Innate expression in CPI treated patients.** Scatterplots showing the adaptive immune expression vs. the innate immune expression in each patient for the CPI treated HMF cohort. The points are coloured by their response to immunotherapy and the plot is stratified by cancer types.

that across cancer types, there was no longer a significant difference in survival. This indicates that basic differences in the balance between adaptive and innate immune activation between genders may be a significant driver of differences in response rates and cancer outcome between men and women.

Currently the standard for stratifying metastatic patients into immunotherapy is to use PDL1 expression. However, some patients with lower levels of PDL1 expression still benefit from the treatment. TMB has been proposed as a predictor for response, and has been used in clinical trials, and while not all tumours with high TMB show response to immunotherapy, an association between TMB and immunotherapy response has been established [42]. TMB is a proxy measure of cancer neoantigens, which is a target of the immune system [43]. This makes neoantigens an obvious candidate for response biomarkers to immunotherapy. However, for the immune system to target cancer cells through neoantigens, T-cells must be present in the environment. We believe the A/I ratio may be relevant to more accurately identify patients who are likely to benefit from CPI therapy. As the results in this study are based on publicly available pan-cancer data from heterogeneous cohorts, independent experimental validation would be necessary to further validate using a ratio of adaptive to innate immune cells for patient stratification.

To ensure that the immune gene expression signal that defines the A/I ratio does indeed originate from the immune cells in the TME and not from the cancer cells themselves, we compared the ranked gene expression from the tumour samples from TCGA, which contain both cancer cells and cells from the surrounding tissue, to the cancer cell line samples from the CCLE project [19], that only contain cancer cells. We found that the expression of immune genes was consistently much lower in the cancer cell lines, median in the bottom 25%, relative to the tumour samples, median in the top 50%. Based on these results, we are confident that the immune signal we observed in our analysis originates from the infiltrating immune cells found within the TME.

Other studies have used a similar approach to identify elements from the immune system, which could be used to make a prediction model for cancer outcome. In a study from 2019, 11 immune regulating genes were used to calculate a risk score. With this, the authors were able to predict 5 year survival with an AUC value of 0.634 for cervical cancer [44]. While the

expression of the genes included in their analysis did not correlate to the expression of the genes used in our analysis, it shows that immune related genes carry prognostic information. It has also been reported that interferon-γ, and interleukin-6 and -10 in advanced melanoma were significantly higher in the patients with response to nivolumab treatment [45]. Immune cells have also been reported to function as predictors for response; NK cells, CD4+ and CD8+ [46]. A study reports that the baseline circulating myeloid-derived suppressor cells (MDSC) correlate with outcome of ipilimumab treatment [47], they find that a low frequency correlates to improved outcome. Another study reported that a subsection of these cells, Polymorphonuclear myeloid-derived suppressor cells, is associated with bad prognosis and resistance to immune checkpoint inhibitor therapy in patients with metastatic melanoma [48]. These studies support that the immune system is the place to look if one wants to find possible predictors for response to immunotherapy.

While we have found a highly significant association between the A/I ratio and patient outcome, both in pan-cancer analysis and in a meta-analysis across cancer types, the A/I ratio was not found to be significantly associated for all cancer types individually. A part of this may be explained by limited power in some cancer types, where there are not enough patients within each group to reach statistical significance, despite a supporting trend. However, undoubtedly cancer-type specific differences also play a role. It is all but certain that the A/I ratio, despite being a systemic marker rather than a cancer-specific marker, is not prognostically relevant for all cancer types and all therapy types. However, to answer this question more accurately, further studies on larger cohorts are required.

With our study we present a more complex interaction between different compartments of the immune system and cancer cells. This presents a case for a more holistic approach to cancer biomarker development, where not only individual molecules, cell types or cancer genomic aberrations are measured and correlated to outcome, but where also the patient's ability to mount an effective immune response to cancer is considered. A key marker here may be the relative infiltration of adaptive to innate immune cells, that may inform on whether the body recognises and is capable of fighting off cancer cells, given appropriate therapy. An advantage of our ratio-based approach is that with a ratio the values are normalised for each individual independently. Thus, while the total activation of the immune system in the tumour microenvironment of individual patients may vary greatly, the ratio may still be informative on an individual level and remain comparable across a cohort of patients.

Overall, our study supports a model where a strong activation of the adaptive immune response relative to the innate immune response in the tumour microenvironment is beneficial to patient outcome. Furthermore, our study provides a potential link between increased cancer-associated mortality among males and a relatively lower ratio of adaptive-to-innate immune cells in the tumour microenvironment. This is consistent with literature indicating gender-specific immune differences, and suggests that biomarker development and drug response predictions must consider gender both in the design and implementation phase. While we found that gender differences were reduced or eliminated when CPI were administered, this must be confirmed in further studies.

## Supporting information

**S1 Fig. Survival across cancer types in TCGA.** a) A forest plot showing the hazard ratio from a multivariate cox proportional hazard regression for progression of cancer for the expression of each of the cell types in the TIL calculation, gender, age, stage and cancer types as covariates. b) A forest plot showing the hazard ratio from a univariate cox proportional hazard regression for progression of cancer. A univariate model was done for each cancer type and for both

genders within the cancer type individually.
(PDF)

**S2 Fig. A/I ratio across TCGA cancer types.** The A/I ratio for 29 cancer types in the TCGA cohort. The cancertypes are ordered by median A/I ratio, female and male patients are represented by red and blue dots respectively, the median for each gender in each cancer type is represented by a horizontal line.
(PDF)

**S3 Fig. Ranked immune gene expression.** The mean ranked expression per cancer type of the 67 immune related genes that the celltype scores are calculated from, for both the cancer cell line data from the CCLE project, and from the tumour samples in TCGA. Low rank = low expression.
(PDF)

**S4 Fig. Survival vs. A/I ratio for TCGA cancer types.** Kaplan-Meier curves showing the 15-year survival for each cancer type within the TCGA cohort, the patients are stratified by gender and A/I ratio.
(PDF)

**S5 Fig. Survival for TCGA cancer types.** Kaplan-Meier curves showing the 10-year survival for TCGA patients with an A/I score above the female median, Male vs. Females. A p-value for the difference in survival is available for each cancer type.
(PDF)

**S6 Fig. Survival vs. A/I ratio for HMF cancer types.** Kaplan-Meier curves showing the five-year survival for each cancer type within the HMF cohort, the patients are stratified by gender and A/I ratio.
(PDF)

**S7 Fig. Survival stratified on gender.** a) Kaplan-Meier curve of 10,158 patients from the TCGA dataset. b) Kaplan-Meier curve of 1746 patients from the HMF dataset. c) Kaplan-Meier curve of 348 patients from the Mariathasan bladder cancer dataset, all CPI treated. d) Kaplan-Meier curve of CPI treated patients from the HMF BLCA dataset. e) Kaplan-Meier curve of CPI treated patients from the HMF LUNG dataset. f) Kaplan-Meier curve of CPI treated patients from the HMF SKCM dataset.
(PDF)

**S8 Fig. Response stratified on gender.** a) Gender stratified response to immunotherapy for the Mariathasan dataset. P-value for difference between gender for each category. b) Gender stratified response to immunotherapy for the HMF dataset, separated by cancer type. P-value for difference between gender for each category.
(PDF)

**S9 Fig.** a) The A/I ratio vs. the immune phenotype for the Mariathasan dataset, separated by gender. P-values are for comparisons between inflamed vs. desert and inflamed vs. excluded. b) Kaplan-Meier curve for survival of TCGA patients with a high or low TIL score. c-e) A/I ratio vs TIL score for the CPI treated patients separated by cancer type, coloured by response category. In the respective cancer types (BLCA: P = 0.003, LUNG: P = 0.15, SKCM: P = 0.006).
(PDF)

**S10 Fig. Survival vs. TIL score for TCGA cancer types.** Kaplan-Meier curves showing the 15-year survival for each cancer type within the TCGA cohort, the patients are stratified by

gender and TIL score.
(PDF)

**S11 Fig. Adaptive vs Innate expression in CPI treated patients.** Scatterplots showing the adaptive immune expression vs. the innate immune expression in each patient for the Mariathasan cohort. The points are coloured by their response to immunotherapy.
(PDF)

**S1 Table. Genes used to define the cell types.** A list of the 67 genes used in the gene expression analysis to define the expression of the cell types used for further analyses [24].
(TXT)

**S2 Table. Result of multivariable Cox proportional hazard model.** Full table of results from the multivariable Cox proportional hazard model from Fig 1A with all covariates.
(TXT)

## Acknowledgments

The results shown here are in part based upon data generated by the TCGA Research Network: https://www.cancer.gov/tcga. This publication and the underlying research are partly facilitated by Hartwig Medical Foundation and the Center for Personalized Cancer Treatment (CPCT) which have generated, analysed and made available data for this research.

## Author Contributions

**Conceptualization:** Johanne Ahrenfeldt, Ditte S. Christensen, Judit Kisistók, Mateo Sokač, Nicolai J. Birkbak.

**Formal analysis:** Johanne Ahrenfeldt, Andreas B. Østergaard.

**Methodology:** Judit Kisistók, Mateo Sokač.

**Resources:** Nicolai J. Birkbak.

**Software:** Johanne Ahrenfeldt, Mateo Sokač.

**Visualization:** Johanne Ahrenfeldt.

**Writing – original draft:** Johanne Ahrenfeldt, Ditte S. Christensen, Nicolai J. Birkbak.

**Writing – review & editing:** Johanne Ahrenfeldt, Nicolai J. Birkbak.

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
