## [Decision Letter · Decision Letter 0]

23 Jun 2022

PONE-D-22-08886The ratio of adaptive to innate immune cells differs between genders and associates with improved prognosis and response to immunotherapyPLOS ONE

Dear Dr. Ahrenfeldt,

Thank you for submitting your manuscript to PLOS ONE. After careful consideration, we feel that it has merit but does not fully meet PLOS ONE’s publication criteria as it currently stands. Therefore, we invite you to submit a revised version of the manuscript that addresses the points raised during the review process.We have based the decision on just one external reviewer. The methodology of the study might only be evaluated by very few scientist working on exactly the same field which has limited the choice of potential reviewers and which has led a significant delay in processing the manuscript. I apologize for it.

We look forward to receiving your revised manuscript.

Kind regards,

Albert Rübben, Ass. Prof., M.D., Ph.D.

Academic Editor

PLOS ONE

Journal Requirements:

2. In your ethics statement in the manuscript and in the online submission form, please provide additional information about the patient records used in your retrospective study. Specifically, please ensure that you have discussed whether all data were fully anonymized before you accessed them.

“The results shown here are in part based upon data generated by the TCGA Research Network: https://www.cancer.gov/tcga. This publication and the underlying research are partly facilitated by Hartwig Medical Foundation and the Center for Personalized Cancer Treatment (CPCT) which have generated, analysed and made available data for this research. NJB is a fellow of the Lundbeck Foundation (R272-2017-4040), and acknowledges funding from Aarhus University Research Foundation (AUFF-E-2018-7-14), and the Novo Nordisk Foundation (NNF21OC0071483).”

“NJB is a fellow of the Lundbeck Foundation (R272-2017-4040), and acknowledges funding from Aarhus University Research Foundation (AUFF-E-2018-7-14), and the Novo Nordisk Foundation (NNF21OC0071483).

Additional Editor Comments:

Although it remains unclear whether the analysis of the RNAseq data might really allow determining the host immune response within a cancer, the sole fact that a pattern of protein expression associated with immune functions demonstrated correlation with the outcome of immunotherapy is of significant importance.

The authors should enter the abbreviation list for the different cancer types within the manuscript and not only in the supplement.

It could be helpful to include a table with the main genes which allow differentiation between innate and adaptive immunity.

In addition, the authors should provide information for all analyzed cancer types on gene expression of proteins involved in immune functions that might be expressed by the cancer itself (for example based on cell culture experiments) and not by infiltrating immune cells (such as IL6 in malignant melanoma), as this might constitute a confounding factor.

The possible expression of these proteins by cancer cells should be discussed within the discussion section.

Reviewers' comments:

Reviewer's Responses to Questions

**Comments to the Author**

1. Is the manuscript technically sound, and do the data support the conclusions?

Reviewer #1: No

2. Has the statistical analysis been performed appropriately and rigorously? 

Reviewer #1: No

3. Have the authors made all data underlying the findings in their manuscript fully available?

Reviewer #1: Yes

4. Is the manuscript presented in an intelligible fashion and written in standard English?

Reviewer #1: Yes

5. Review Comments to the Author

Reviewer #1: The authors investigate an important question in immune-oncology: does the balance between innate and adaptive immunity hold important prognostic/predictive information? They investigate this questions in 3 publicly-available datasets, including TCGA. The approach of looking at adaptive /immune ratio is interesting and worthwhile, and the analyses are rich. However, there are flaws in the statistical analyses that make many of the paper’s key results unsupported. If the analyses are corrected, presumably causing many key results to change, then this manuscript could be a worthy addition to the literature.

Major concern 1:

Many of the study’s key statistical models do not consider cancer type, leaving them subject to confounding and erroneous results. These models must be corrected for the paper’s conclusions to be believed. The minor concerns below expand on this topic.

Major concern 2:

The standard model of immune-oncology would hold that, “adaptive immune cells are helpful for survival.” With its focus on adaptive/innate ratio, this paper claims that innate cells hold additional information. But this claim is never directly evaluated. (There is a multivariate analysis of A/I ratio and total TILs, but I believe total TILs includes both innate and adaptive populations.) To see whether innate cells have any prognostic information to add, multivariate models should be run with both halves of the A/I score. Then we could see whether adaptive cells are the whole story, or whether innate cells have a countervailing effect. Unless this analysis is done, I’m left to wonder whether A/I ratio is really only meaningful insofar as it provides an oblique readout of total adaptive immunity.

More minor concerns are below:

The approach of contrasting innate and adaptive scores is interesting and sound. (Not a concern.)

Line 72: “macrophages are the most abundant” – add a reference please.

The text needs some light editing for spelling, grammar, and conciseness.

Line 106: only PD, PR and CR were used. Were Stable Disease cases omitted then? If so, why? Without a justification, this looks like cherry-picking the data.

117: “a linear scaling was performed” – please give the full details of this operation, and the motivation. Because the cell type scores are log-scale, some scaling approaches might lead to strange interpretations, while simply mean-centering would retain the log-scale interpretation of the scores.

Figure 1a: Fitting a single model to all cancer types doesn’t make much sense. In particular, I’m worried that cancer type acts as a powerful confounder in the analysis – e.g. melanoma is both highly infiltrated and deadly, confounding the relationship between immune abundance and survival. Secondly, it seems hard to justify the assumption that immune cells have the same survival implications in each cancer type. (For example, in glioblastoma, I believe high immune abundance predicts poor survival). To address these concerns, please run this analysis separately for each cancer type and report those results instead. And ideally, BRCA would be split into HR+ vs. TNBC, and COAD would be split into MSI-high vs. MSS.

Figure 1a: in addition, please expand your regressions to adjust for other variables relevant to survival, e.g. tumor grade, stage, sex, age, etc…

The TCGA datasets DLBC and LAML (and possibly THYM) are tricky for immune decomposition, as these immune-driven cancers express many immune marker genes. I recommend removing them to ensure they don’t contaminate the more reliable results from solid tumors.

Please define how “total TIL infiltration” is calculated.

Figure 1d: a forest plot might show these results to better effect. For example, is the COAD hazard ratio for males significantly different than it is for females? A forest plot would show this, while the volcano plot cannot.

Figure 1e: can you label the low outlier cancer type? That seems interesting.

“…the specific ratio of adaptive 161 to innate immune cells is more relevant to disease outcome than total TIL infiltration.” This claim is both important and novel, and so it deserves to be backed up more thoroughly.

Very important: the regression of survival on A/I and TILs should be run separately for each cancer type; otherwise cancer type is a confounder.

Figure 2a: same concern: a model combining all cancer types together is probably invalid. Please re-run this model separately for each cancer type.

Lines 167-171: This argument ignores how different cancer types have different incidence in males and females. E.g. breast cancer, the biggest TCGA dataset, occurs mainly in females. Thus the argument does not support the conclusion that “This suggests that some of the 171 gender difference in survival can be predominantly explained by the increased A/I ratio in females.” Instead, you could model survival vs. sex, then again vs. sex + A/I ratio. The change in sex’s effect size between the models would get at how much of sex effects are explained by A/I. (As with all the other models, this analysis would have to be performed per cancer type.)

Line 178, “This suggests that a high A/I ratio may lead to a lower frequency of patients progressing to 179 metastatic disease.” That’s making too strong a conclusion from the available data – there are all sorts of reasons different study populations could differ. Please remove this claim or support it more rigorously.

Figure 2b: please clarify which cohort this analysis is from.

Figure 2b: please stratify by cancer type

Figure S5b: please stratify by cancer type

Line 279-281: “This suggests that immunotherapy may be relatively more effective in males than females, 280 with males achieving greater benefit from immunotherapy compared to chemotherapy or targeted 281 therapies.” This is phrased too strongly, unless you can back it up with some statistics (including confidence intervals).

6. PLOS authors have the option to publish the peer review history of their article (what does this mean?). If published, this will include your full peer review and any attached files.

Reviewer #1: No

---

## [Author Response · Author response to Decision Letter 0]

14 Sep 2022

Dear reviewer,

First of all, we would like to thank you for taking the time to revise our paper. We greatly appreciate this. We have found your comments useful, and followed them to make the paper even better. Below is a point by point list of our responses to each of them. 

Best Regards,

Nicolai J. Birkbak & Johanne Ahrenfeldt

Reviewer #1: The authors investigate an important question in immune-oncology: does the balance between innate and adaptive immunity hold important prognostic/predictive information? They investigate this questions in 3 publicly-available datasets, including TCGA. The approach of looking at adaptive /immune ratio is interesting and worthwhile, and the analyses are rich. However, there are flaws in the statistical analyses that make many of the paper’s key results unsupported. If the analyses are corrected, presumably causing many key results to change, then this manuscript could be a worthy addition to the literature.

Major concern 1:

Many of the study’s key statistical models do not consider cancer type, leaving them subject to confounding and erroneous results. These models must be corrected for the paper’s conclusions to be believed. The minor concerns below expand on this topic.

We apologise for this oversight. We have now performed all models on each cancer type individually or with the cancer types as covariates (particularly note the updated supplementary figure 1). We are happy to report that this has not in any major way changed our results, in most cases the A/I ratio remains significantly associated with outcome. In some cancer types, the association is not significant or less significant, but here the trend mostly remains, and some of this is likely a power issue.

Major concern 2:

The standard model of immune-oncology would hold that, “adaptive immune cells are helpful for survival.” With its focus on adaptive/innate ratio, this paper claims that innate cells hold additional information. But this claim is never directly evaluated. (There is a multivariate analysis of A/I ratio and total TILs, but I believe total TILs includes both innate and adaptive populations.) To see whether innate cells have any prognostic information to add, multivariate models should be run with both halves of the A/I score. Then we could see whether adaptive cells are the whole story, or whether innate cells have a countervailing effect. Unless this analysis is done, I’m left to wonder whether A/I ratio is really only meaningful insofar as it provides an oblique readout of total adaptive immunity.

We appreciate this insightful comment from the reviewer. This is indeed a very good point. Based on this suggestion, we have made a multivariate model where we test for both adaptive and innate immune cells as covariates This has been done with and without cancer types as covariates in the model. Omitting cancer types, we see a very striking result, where both innate and adaptive immune cells are significantly associated with outcome in opposite directions (Adaptive HR = 0.016, P = 9.20*10^-7, Innate HR = 80.9, P= 1.94*10^-6). Including cancer types in the model, we observe the same signal, albeit weaker (Adaptive HR = 0.071, P = 0.0014, Innate HR = 1.042, P= 0.57). 

We have included this analysis in the manuscript:

“To evaluate if the two compartments of the immune system do indeed pull in opposite directions when it comes to patient outcome, we used a multivariate model, where the innate and the adaptive values were covariates together with age, stage and gender. This was done both pan-cancer and including the cancer types as covariates. We find that a high adaptive component is significantly associated better survival (pan-cancer: HR = 0.016, P = 9.20*10-7, cancer informed: HR = 0.071, P = 0.0014,) where as a high innate component is associated with an poor prognosis, although only significantly in pan-cancer (pan-cancer: HR = 80.9, P= 1.94*10-6, cancer informed: HR = 1.042, P= 0.57).”

More minor concerns are below:

The approach of contrasting innate and adaptive scores is interesting and sound. (Not a concern.)

Line 72: “macrophages are the most abundant” – add a reference please.

Our apologies, the reference for this was inserted further below. We have now moved the reference up to the end of this sentence. Furthermore the sentence was changed to “Tumour associated macrophages are one of the most abundant cells in the TME (Singh et al. 2019)” 

The text needs some light editing for spelling, grammar, and conciseness.

Line 106: only PD, PR and CR were used. Were Stable Disease cases omitted then? If so, why? Without a justification, this looks like cherry-picking the data.

Thank you for pointing this out. Not including our justification was an oversight. We have now added our reasons behind this decision. Stable disease is specifically not included in the analysis, as it is not clearly associated with either response or disease progression. Patients with stable disease may have extended survival can be considered as experiencing clinical benefit from treatment, even if the tumour does not shrink following therapy. To make this clear in the manuscript, we have added the following to the methods:

“Patients with stable disease (SD) were not included in the analysis, as the interpretation of SD is not clearly defined as good or poor outcome. Indeed, it can be both a sign that the therapy works and contains tumour growth, or it can be a sign that the therapy has no effect but the tumour size remains unchanged due to stagnated growth”

117: “a linear scaling was performed” – please give the full details of this operation, and the motivation. Because the cell type scores are log-scale, some scaling approaches might lead to strange interpretations, while simply mean-centering would retain the log-scale interpretation of the scores.

We wanted to have all of the values between 0 and 1, for easy interpretation of the calculation. This makes the value for all cell types comparable, and we can then sum them up and use their mean. All values were un-logged prior to scaling. For each cell type the maximum and minimum value was found, and the scaled value was calculated as scaled = (unscaled - minimum ) / (maximum - minimum). 

We have changed the text in the methods to describe this in more detail: 

“A linear scaling of the expression values for each cell type was performed as follows, first the values were reverse log-transformed, and then the values within each cell type were linearly scaled to values between 0 and 1, with this equation: scaled_celltypen = (celltypen - celltypemin ) / (celltypemax - celltypemin), and then a mean scaled expression a score for each group (adaptive and innate) was calculated per sample as the mean scaled value for the cell types within the group per sample, whereafter the A/I ratio was determined by dividing the adaptive with the innate score.”

Figure 1a: Fitting a single model to all cancer types doesn’t make much sense. In particular, I’m worried that cancer type acts as a powerful confounder in the analysis – e.g. melanoma is both highly infiltrated and deadly, confounding the relationship between immune abundance and survival. Secondly, it seems hard to justify the assumption that immune cells have the same survival implications in each cancer type. (For example, in glioblastoma, I believe high immune abundance predicts poor survival). To address these concerns, please run this analysis separately for each cancer type and report those results instead. And ideally, BRCA would be split into HR+ vs. TNBC, and COAD would be split into MSI-high vs. MSS.

When we modelled on single cancer types, or added cancer types as a covariate, most cell types were not individually significant, occasionally except for Macrophages, Mast cells and NK cells. Thus, to leverage the power of multiple cell types with correlated abundance, we decided to investigate the ratio of adaptive to innate immune cells. To make this more clear, we have now added a forest plot from the model with cancer types as covariates to Supplementary figure 1a, and expanded on our reasoning behind the AI-ratio in the explanatory text:

“We then fitted a multivariable Cox proportional hazard model to the progression free interval, including all immune cell types and gender, age and tumour stage as covariates (Figure 1A omitting age, stage and gender from the visualisation. Full results with all covariates listed in Table S2). Of 14 immune cell types, 8 showed a significant association with outcome, four with improved survival, four with worse survival. Overall, we observed that adaptive immune cells associated with a lower risk of relapse (CD8 T-cells, CD45, T-cells, Th1 cells, Treg), while innate immune cells was associated with a higher risk of relapse (Dendritic cells, Macrophages, Natural killer cells) (Figure 1a). When we performed the same analysis including cancer types as covariates, the same overall pattern was observed (Figure S1a). To further evaluate how the two compartments of the immune system associate with patient outcome in opposite directions, we divided the cell types based on which of the two major immune components they belong to and calculated a value for each component. We then performed a multivariate model including the innate and the adaptive values together with age, stage and gender. This was done separately pan-cancer and with cancer type as covariates. We found that a high adaptive component is significantly associated with improved survival (pan-cancer: HR = 0.016, P = 9.20*10-7, cancer informed: HR = 0.071, P = 0.0014,) whereas a high innate component is associated with an poor prognosis, although only significantly in pan-cancer (pan-cancer: HR = 80.9, P= 1.94*10-6, cancer informed: HR = 1.042, P= 0.57).”

Additionally, while we agree that splitting BRCA and COAD into subtypes would be optimal, this also reduces the power of the cohorts.

Figure 1a: in addition, please expand your regressions to adjust for other variables relevant to survival, e.g. tumor grade, stage, sex, age, etc…

The model used to produce figure 1a, do already contain age, sex and stage. To improve clarity, we have added more detail to the text where we explain this:

“We then fitted a multivariable Cox proportional hazard model to the progression free interval, including all immune cell types and gender, age and tumour stage as covariates (Figure 1A omitting age, stage and gender from the visualisation. Full results with all covariates listed in supplementary table 2)

The TCGA datasets DLBC and LAML (and possibly THYM) are tricky for immune decomposition, as these immune-driven cancers express many immune marker genes. I recommend removing them to ensure they don’t contaminate the more reliable results from solid tumors.

We thank the reviewer for this insightful comment. We agree that is a valid point, and have removed these three cancer types from the analysis. 

Please define how “total TIL infiltration” is calculated.

We apologise for the lack of clarity. We have expanded the text describing how the total TIL infiltration is calculated in the methods section, pasted here below:

“Tumour immune cell decomposition was performed using the score defined by Danaher and colleagues21 based on whole tumour RNAseq data, implemented as described22. We used a defined list of genes from Danaher(Danaher et al. 2017) to define the expression of immune cell types, and the mean of the cell types described in the paper was then used as the total TIL score” 

Figure 1d: a forest plot might show these results to better effect. For example, is the COAD hazard ratio for males significantly different than it is for females? A forest plot would show this, while the volcano plot cannot.

We thank the reviewer for this suggestion. While we specifically chose a volcano plot as we believe it is a more illustrative method to show the effect of gender, we have now also made a forest plot version. Both methods of course show the same results, but we are respectfully of the opinion that the volcano plot is more illustrative. Nevertheless, we have included the forest plot as Supplementary Figure 1b.

Figure 1e: can you label the low outlier cancer type? That seems interesting.

We agree this is of interest and thank the reviewer for the suggestion. We have added labels to the outliers.

“…the specific ratio of adaptive 161 to innate immune cells is more relevant to disease outcome than total TIL infiltration.” This claim is both important and novel, and so it deserves to be backed up more thoroughly.

Very important: the regression of survival on A/I and TILs should be run separately for each cancer type; otherwise cancer type is a confounder.

We thank the reviewer for the nice comments and the suggestion, we have performed the analysis, and we have changed the paragraph in the paper so reflect the added analyses:

“While the total TIL score was significant in univariate analysis (HR = 0.93, p = 2.95x10^-07), it was not significant in the multivariate analysis (TIL HR = 1.032, P = 0.06114). However when we run this model with cancetype as a covariate, both terms are associated with significantly better outcome (AI ratio: HR = 0.92 , P = 0.000917, TIL: HR = 0.94 , P = 0.002245), indicating that the specific ratio of adaptive to innate immune cells is slightly more relevant to disease outcome than total TIL infiltration, but not for all cancer types.” 

Figure 2a: same concern: a model combining all cancer types together is probably invalid. Please re-run this model separately for each cancer type.

Thank you for this comment, we agree that the cancer types are important to incorporate in the analysis. We have now performed the analysis, the results are in a supplementary figure 4.

Lines 167-171: This argument ignores how different cancer types have different incidence in males and females. E.g. breast cancer, the biggest TCGA dataset, occurs mainly in females. Thus the argument does not support the conclusion that “This suggests that some of the 171 gender difference in survival can be predominantly explained by the increased A/I ratio in females.” Instead, you could model survival vs. sex, then again vs. sex + A/I ratio. The change in sex’s effect size between the models would get at how much of sex effects are explained by A/I. (As with all the other models, this analysis would have to be performed per cancer type.)

This model has been performed for each cancer type individually, and cancer types where there are mainly female patients have been excluded from this analysis, so Breast, Cervix and Ovarian cancer are not part of this analysis. Neither is Prostate cancer, as there are no females to compare with. 

We have followed the suggestion to model survival vs. gender and compared it to survival vs. gender + A/I ratio. And we did this with the cancer types as covariates. The results show that the model with our A/I ratio is a significantly better model than only gender. This has been added to the results: 

“To investigate if the differences in survival were solely based on gender, we performed two cox proportional hazard models, one analysing survival relative to gender, and one analysing survival relative to gender and the A/I ratio. We then compared the performance of the models using a likelihood ratio test. Based on this analysis, we found that the model including the A/I ratio term significantly out-performed the simpler model including only gender (P = 4.45 * 10-9).”

Line 178, “This suggests that a high A/I ratio may lead to a lower frequency of patients progressing to 179 metastatic disease.” That’s making too strong a conclusion from the available data – there are all sorts of reasons different study populations could differ. Please remove this claim or support it more rigorously.

We apologise the strong language, we have rephrased this section as:

“Taken together, this suggests that a high A/I ratio may be one of the factors that contribute to a lower frequency of patients progressing to metastatic disease.”

Figure 2b: please clarify which cohort this analysis is from.

We apologise for the oversight, we have improved the text describing Figure 2 as:

“Next, we performed a survival analysis on the metastatic HMF cohort and found that both male and female patients had improved overall survival if their A/I ratio was above median (figure 2b)”

Additionally, the cohort is mentioned on the figure itself (“HMF”), and in the figure legend

Figure 2b: please stratify by cancer type

Apologies, we have made the figure for each individual cancertype, these have been added as supplementary figure 6. 

Figure S5b: please stratify by cancer type

We have stratified based on cancertype, and added this as supplementary figure 10.

Line 279-281: “This suggests that immunotherapy may be relatively more effective in males than females, 280 with males achieving greater benefit from immunotherapy compared to chemotherapy or targeted 281 therapies.” This is phrased too strongly, unless you can back it up with some statistics (including confidence intervals).

Thank you for pointing this out, we have tried and failed to phrase it less strongly or to find a way to show this statistically, but instead we have ended up removing this sentence, as we do not have sufficient data to back it up. 

Additional Editor Comments:

Although it remains unclear whether the analysis of the RNAseq data might really allow determining the host immune response within a cancer, the sole fact that a pattern of protein expression associated with immune functions demonstrated correlation with the outcome of immunotherapy is of significant importance.

The authors should enter the abbreviation list for the different cancer types within the manuscript and not only in the supplement.

We have added a list of abbreviations as Table 1. 

It could be helpful to include a table with the main genes which allow differentiation between innate and adaptive immunity.

Thank you for this input, we have added the genes as Supplementary Table 1.

In addition, the authors should provide information for all analyzed cancer types on gene expression of proteins involved in immune functions that might be expressed by the cancer itself (for example based on cell culture experiments) and not by infiltrating immune cells (such as IL6 in malignant melanoma), as this might constitute a confounding factor.

The possible expression of these proteins by cancer cells should be discussed within the discussion section.

Thank you for this very good point, we agree that this does add a layer of nuance to the discussion. We therefore used data from the CCLE (Cancer Cell Line Expression) project downloaded from DepMap (DepMap, Broad (2022): DepMap 22Q2 Public. figshare. Dataset. https://doi.org/10.6084/m9.figshare.19700056.v2) to evaluate the expression of immune genes. For each sample the genes were ranked based on expression values (low rank = low expression), and for each cancer type, we calculated the mean rank score per gene. We did the same for TCGA, and in supplementary figure 3 we now show the values for the cancer types where data from both datasets were available. 

We have added the following to the results:

“To confirm that the expression of immune related genes are in fact originating from the TME and not from the cancer cells we explored the expression of the individual immune genes in the Cancer Cell Line Encyclopedia (CCLE)29, a dataset of cancer cell lines (thus devoid of any infiltrating immune cells). Here we observed that the ranked expression of the immune genes were low for all cancer cell lines, except for cell lines originating from leukaemia and lymphoma, both cancers of the immune system. When we compared the ranked expression of cancer cell lines to TCGA tumour samples of matched tissue, we observed significantly higher ranks for all tumours (Figure S3), indicating that the observed immune signal is indeed originating from infiltrating immune cells, and is not from cancer cells expressing immune-related genes.”

And the following to the discussion: 

“To ensure that the immune gene expression signal we see from the A/I ratio does indeed come from the immune cells in the TME and not from the cancer itself, we compared the ranked gene expression from the tumour samples from TCGA, which contain both cancer cells and cells from the surrounding tissue, to the cancer cell line samples from the CCLE project29, that only contain cancer cells. We found that the expression of immune genes are consistently much lower in the cancer cells, median in the bottom 25 %, than in the mixed cells, median in the top 50 %. Based on these results, we are confident that the immune signal we observe in our data analysis originates from the infiltrating immune cells found within the TME.”

---

## [Decision Letter · Decision Letter 1]

17 Oct 2022

PONE-D-22-08886R1The ratio of adaptive to innate immune cells differs between genders and associates with improved prognosis and response to immunotherapyPLOS ONE

Dear Dr. Ahrenfeldt,

Thank you for submitting your manuscript to PLOS ONE. After careful consideration, we feel that it has merit but does not fully meet PLOS ONE’s publication criteria as it currently stands. Therefore, we invite you to submit a revised version of the manuscript that addresses the points raised during the review process.

We look forward to receiving your revised manuscript.

Kind regards,

Albert Rübben, Ass. Prof., M.D., Ph.D.

Academic Editor

PLOS ONE

Reviewers' comments:

Reviewer's Responses to Questions

**Comments to the Author**

1. If the authors have adequately addressed your comments raised in a previous round of review and you feel that this manuscript is now acceptable for publication, you may indicate that here to bypass the “Comments to the Author” section, enter your conflict of interest statement in the “Confidential to Editor” section, and submit your "Accept" recommendation.

Reviewer #1: (No Response)

2. Is the manuscript technically sound, and do the data support the conclusions?

Reviewer #1: No

3. Has the statistical analysis been performed appropriately and rigorously? 

Reviewer #1: No

4. Have the authors made all data underlying the findings in their manuscript fully available?

Reviewer #1: (No Response)

5. Is the manuscript presented in an intelligible fashion and written in standard English?

Reviewer #1: Yes

6. Review Comments to the Author

Reviewer #1: Thank you for addressing my comments; I agree the manuscript is stronger now. It’s got many interesting results, presented well.

Before publication, I have to insist: the manuscript should not include any results from pan-cancer analyses ignoring cancer type. Because of the very strong confounding by tumor type, the results of these analyses are misleading. The detailed comments call out results of this nature. To be clear: I consider all these detailed comments to be essential and will not support publication unless they are addressed.

Detailed comments:

- Thanks for clarifying the scaling approach. I have no concerns about the described approach.

- This claim in the abstract is from a pan-cancer / cancer-type-blind analysis and must be replaced: “Pan-cancer analysis of primary tumour samples from TCGA showed improved progression free survival in 30 patients with an A/I ratio above median (P < 0.0001).”

- This claim in the abstract is from a pan-cancer / cancer-type-blind analysis and must be replaced: For patients with metastatic disease, we found that 33 responders to immunotherapy have a significantly higher A/I ratio than non-responders in HMF (P = 0.036).

- Figure 1a still reports results ignoring cancer type. Please remove these results entirely and replace them with the results of Figure S1.

- The COAD samples still haven’t been split into MSI high and MSS. Because these subtypes are profoundly immunologically different, ignoring them makes it hard to interpret your results for this cancer type. E.g. I see males and females have very different results for COAD in Figure S1A, but I don’t know if that’s just confounding due to MSI status. The TCGA clinical metadata will have this field. I know that stratifying further reduces sample size, but if the alternative to an underpowered analysis is a confounded analysis, then the underpowered analysis is the right choice. This same comment applies to TNBC BRCA, but the effect these is less dramatic.

- “We found that a high adaptive component is significantly associated with improved survival (pan-cancer: HR = 0.016, P = 9.20*10-7 173 , cancer informed: HR = 0.071, P = 0.0014,) whereas a high 174 innate component is associated with an poor prognosis, although only significantly in pan-cancer (pancancer: HR = 80.9, P= 1.94*10-6 175 , cancer informed: HR = 1.042, P= 0.57)” -> Please remove the result from the unadjusted analysis.

- Thank you for the analysis including AI ratio and total TILs. I think this statement is not supported by the printed results: “indicating that the specific ratio of adaptive to innate immune cells is slightly more relevant to disease outcome than total TIL infiltration, but not for all cancer types”. Please either remove it or give a p-value for the difference between the 0.92 and 0.94 hazard ratios.

- “We observed that the AI ratio remained highly significant (AI ratio: HR = 0.77 , P < 2x10^-16). While the total TIL score was significant in univariate analysis (HR = 0.93, p = 2.95x10-7 215 ), it was not 216 significant in the multivariate analysis (TIL HR = 1.03, P = 0.061).” -> Please remove any reference to pan-cancer analyses ignoring tumor type.

- Figure 2a: please remove, since it ignores cancer type. Moving the results of S5 to a main figure would be appropriate (or at least a summary of them, e.g. a forest plot).

- Figure 2b: please remove, since it ignores cancer type.

- “To investigate if the differences in survival were solely based on gender, we performed two cox proportional hazard models, one analysing survival relative to gender, and one analysing survival relative to gender and the A/I ratio. We then compared the performance of the models using a likelihood ratio test. Based on this analysis, we found that the model including the A/I ratio term significantly out-performed the simpler model including only gender (P = 4.45 * 10-9 237 )” -> I’m reading this as ignoring cancer type. Please remove, or at a minimum adjust for cancer type.

- Figure S7: Same as above: please remove or stratify by cancer type. (I believe the Mariathasan dataset is all bladder cancer; if so, then its results can be kept.)

- Line 280: “For both cohorts we found no significant difference in 281 survival of male and and female patients treated by CPI” -> the HMF results should be stratified by cancer type.

- Figure 3b,d,f: please remove or stratify by cancer type.

- “This supports our findings that metastatic melanoma with a poor response to CPI have a lower A/I ratio, indicating that they have a relatively high expression of the innate immune system.” -> This is a rather indirect inference. You should either remove this statement or back it up with an analysis of the innate immune score in the relevant samples.

- It looks like your analyses found that innate abundance wasn’t informative when considered alongside adaptive abundance. At a minimum, please discuss the implications of this in the Discussion. Should we just be looking at adaptive abundance alone, or does your work provide additional reasons to look at A/I ratio?

7. PLOS authors have the option to publish the peer review history of their article (what does this mean?). If published, this will include your full peer review and any attached files.

Reviewer #1: No

---

## [Author Response · Author response to Decision Letter 1]

6 Dec 2022

Response to reviewers is uploaded as file and inserted below. 

Reviewer #1: Thank you for addressing my comments; I agree the manuscript is stronger now. It’s got many interesting results, presented well.

Before publication, I have to insist: the manuscript should not include any results from pan-cancer analyses ignoring cancer type. Because of the very strong confounding by tumor type, the results of these analyses are misleading. The detailed comments call out results of this nature. To be clear: I consider all these detailed comments to be essential and will not support publication unless they are addressed.

We appreciate the reviewers excellent comments, particularly we note the reviewer agrees with us that the manuscript has been significantly improved following the initial review and with incorporation of the reviewers excellent suggestions into the text and figure layout.

However, we politely disagree with the firm notion that pan-cancer analysis is not a valid scientific approach. Cancer is inherently a genomic disease inflicted upon host cells, all sharing the *exact* same genome initially. The site of origin is critical for the clinical trajectory of the disease, but different cancer types are not fundamentally different diseases, as may be the case for diseases caused by external pathogens, viruses and bacteria. Indeed, pan-cancer analysis may reveal common characteristics in the body’s natural defenses against cancer, but also differences likely dictated by the tissue microenvironments and cellular programming depending on the tissue of origin.

Indeed, pan-cancer analysis is not novel. In the scientific literature, pan-cancer analysis is commonly found already. Eg., in this manuscript by Combes and colleagues, published in Cell earlier this year, the authors deployed a similar type of analysis, as shown in figure 3b:

https://www.sciencedirect.com/science/article/pii/S0092867421014264

And for further examples:

Figure 3a:

https://cancerci.biomedcentral.com/articles/10.1186/s12935-021-02266-3

Figur 8;

https://www.nature.com/articles/s41389-019-0121-7

Figur 1:

https://www.mdpi.com/2072-6694/11/10/1562/htm

We accept that the reviewer may disagree on this approach, however in our humble opinion that does not invalidate the analysis but can be presented as a weakness. To alleviate this concern, we have now included both pan-cancer and cancer-type specific analysis throughout the manuscript, and a specific section in the discussion:

“While we have found a highly significant association between the A/I ratio and patient outcome , both in pan-cancer analysis and in a meta-analysis across cancer types, the A/I ratio was not found to be significantly associated for all cancer types individually. A part of this may be explained by limited power in some cancer types, where there are not enough patients within each group to reach statistical significance, despite a supporting trend. However, undoubtedly cancer-type specific differences also play a role. It is all but certain that the A/I ratio, despite being a systemic marker rather than a cancer-specific marker, is not prognostically relevant for all cancer types and all therapy types. However, to answer this question properly, further studies on larger cohorts are required.

Furthermore, we have added to or changed most manuscript figures to make the cancertype specific results more visible:

Figure 1, updated with COAD MSS & MSI subtypes

Figure 2, completely redesigned with new panels b, c, e, & f to show cancer type specific results

Figure 3, previous figure 2d, reassigned to its own figure, showing cancer type specific results

Figure 4, redesigned previous figure 3, showing cancer type specific results

Figure 5, new figure, showing cancer type specific results comparing adaptive to innate scores

Supplementary figure 4, added COAD MSS and COAD MSI

Supplementary figure 5, added COAD MSS and COAD MSI

Supplementary figure 7, added cancer type to HMF

Supplementary figure 10, added COAD MSS and COAD MSI

Supplementary figure 11, new figure comparing adaptive to innate scores in Mariathasan BLCA

While the reviewer was quite firm in their arguments, we hope the above may sufficiently alleviate their concerns so that this project may now be shared with a wider audience.

All considered, we are very appreciative of the time and effort the reviewer has placed on our manuscript, and there is no question that with the extensive revisions performed during this process the resulting manuscript and the presented analysis has been thoroughly improved. We hope that with this last round of major revisions, we may convince the reviewer that the manuscript is now in a state sufficient for publication.

Detailed comments:

- Thanks for clarifying the scaling approach. I have no concerns about the described approach.

We thank the reviewer for the comment

- This claim in the abstract is from a pan-cancer / cancer-type-blind analysis and must be replaced: 

“Pan-cancer analysis of primary tumour samples from TCGA showed improved progression free survival in 30 patients with an A/I ratio above median (P < 0.0001).”

We have removed this sentence from the abstract, and replaced it with a meta-analysis that considers cancer types

“A meta-analysis of 32 cancer types from TCGA overall showed improved progression free survival in patients with an A/I ratio above median (Hazard ratio (HR) females 0.73, HR males 0.86, P < 0.05)”. 

- This claim in the abstract is from a pan-cancer / cancer-type-blind analysis and must be replaced: For patients with metastatic disease, we found that 33 responders to immunotherapy have a significantly higher A/I ratio than non-responders in HMF (P = 0.036).

We respectfully disagree with this comment, as here we are specifically investigating IO response within a limited group of patients. The question is relevant to ask across all samples, and the text in the main body of the manuscript goes into detail with the individual cancer types where the observation still stands. Hence, in our humble opinion, this is not misleading.

- Figure 1a still reports results ignoring cancer type. Please remove these results entirely and replace them with the results of Figure S1.

The concept behind this manuscript is that the type of immune cell infiltration matters. Figure 1A demonstrates the logical reasoning behind the project, and serves to illustrate which cell types are part of each group. The take-away message here is that some immune cell types are associated with improved outcome, others vice-versa. For full transparency, Figure S1 shows the same analysis with cancer type, here, the HRs demonstrate the same trend, though prognostically, this is unsurprisingly often not significant given the strong prognostic effect of primary cancer type, which we also clearly describe as such in the text. Overall, based on this analysis, we found that, pan-cancer, there was a tendency for the cell types to behave differently depending on which immune cell group they belonged to, compared to survival. We also found that it was different cells within the groups for different cancer types that would be significantly associated with outcome. This is illustrated very poorly by the plot including cancer types, Figure S1A, but remarkably well on the plot without them. Therefore, as this is a discovery analysis, we will argue that it is reasonable to keep the current figure, and politely disagree with the reviewers’ firm opposition to this concept. However we have altered the text to ensure the reader is aware of the purpose of the analysis and the strong association of cancer type and outcome.

“When we performed the same analysis including cancer types as covariates, the same overall pattern was observed with regard to the direction of association of the individual immune cell types, although unsurprisingly cancer type was by far the most significant covariates relative to outcome reflecting established cancer-type specific prognosis (Figure S1a).”

- The COAD samples still haven’t been split into MSI high and MSS. Because these subtypes are profoundly immunologically different, ignoring them makes it hard to interpret your results for this cancer type. E.g. I see males and females have very different results for COAD in Figure S1A, but I don’t know if that’s just confounding due to MSI status. The TCGA clinical metadata will have this field. I know that stratifying further reduces sample size, but if the alternative to an underpowered analysis is a confounded analysis, then the underpowered analysis is the right choice. This same comment applies to TNBC BRCA, but the effect these is less dramatic.

We acknowledge that the MSI status of the COAD samples is important, and we have now added the status to the TCGA patients, and rerun all the plots. What we see most clearly is that the MSS COAD patients have by far the largest gain from a high A/I ratio, and then unlabeled COAD - but still only for the male patients. For the female COAD patients none of the three groups are significant. 

We have added the new results in the manuscript and added the division of the COAD patients in the methods section:

Methods:

“Cancer type abbreviations are found in Table 1. Information regarding MSI status [16] in colon cancer was used to split the COAD patients into COAD MSI, COAD MSS and COAD, the latter for the patients where the information was not available. ”

Results:

“We observed that a higher A/I ratio significantly associated with improved outcome in 12 cancer types (COAD, COAD MSS, HNSC, BLCA, CESC, MESO, UCEC, BRCA, CHOL, LIHC, LUAD & LUSC), supporting the known role of the adaptive immune system in combating cancer[32]. Interestingly, for COAD, COAD MSS, HNSC, LIHC, LUAD and LUSC only males showed a significant association, while for MESO, only females.”

- “We found that a high adaptive component is significantly associated with improved survival (pan-cancer: HR = 0.016, P = 9.20*10-7 173 , cancer informed: HR = 0.071, P = 0.0014,) whereas a high 174 innate component is associated with an poor prognosis, although only significantly in pan-cancer (pancancer: HR = 80.9, P= 1.94*10-6 175 , cancer informed: HR = 1.042, P= 0.57)” -> Please remove the result from the unadjusted analysis.

Here we clearly show the results for the cancer informed model, as well as the pan-cancer model, thus we politely insist that it is not unreasonable to keep the pan-cancer results. 

- Thank you for the analysis including AI ratio and total TILs. I think this statement is not supported by the printed results: “indicating that the specific ratio of adaptive to innate immune cells is slightly more relevant to disease outcome than total TIL infiltration, but not for all cancer types”. Please either remove it or give a p-value for the difference between the 0.92 and 0.94 hazard ratios.

We apologise for our poor phrasing and thank the reviewer for bringing this to our attention. As both terms are significantly associated with outcome, we have changed the text as indicated below:

“When we included cancer type as a covariate in the model, both terms remained significantly associated improved outcome (AI ratio: HR = 0.92 , P = 0.000988, TIL: HR = 0.94 , P = 0.000792), indicating that both the specific ratio of adaptive to innate immune cells and the total amount of immune cells are independently associated with outcome.”

- “We observed that the AI ratio remained highly significant (AI ratio: HR = 0.77 , P < 2x10^-16). While the total TIL score was significant in univariate analysis (HR = 0.93, p = 2.95x10-7 215 ), it was not 216 significant in the multivariate analysis (TIL HR = 1.03, P = 0.061).” -> Please remove any reference to pan-cancer analyses ignoring tumor type.

Here we clearly show both cancer-informed and pan-cancer, thus we politely disagree with the reviewers position.

- Figure 2a: please remove, since it ignores cancer type. Moving the results of S5 to a main figure would be appropriate (or at least a summary of them, e.g. a forest plot).

- Figure 2b: please remove, since it ignores cancer type.

We have added forest plots for both TCGA and HMF, as 2b-c and 2e-f, respectively. To these we have added a meta analysis, to show that overall an A/I ratio above median leads to an improved survival. The text regarding these plots have been changed to: 

“When we performed survival analysis on the combined TCGA cohort including all patients, we found that both female and male patients with an A/I ratio above median had significantly improved overall survival relative to patients with an A/I ratio below median (Figure 2a). We performed the same analysis on the individual cancer types, and found that 7/30 cancer types (BRCA, CESC, HNSC, LICH, OV, SKCM and UCEC) showed significantly improved outcome with an A/I ratio above median, while 2/30 (LGG and UVM) showed the opposite (Figure S4). Based on these results, we performed a metaanalysis which take all cancer types into account, on male and female patients separately. Here, we observed that an A/I ratio above median associated with improved outcome in both male and female patients, but with a stronger association in females (HR females 0.73, HR males 0.86, P < 0.05, Figure 2b-c).”

“Initially, we performed a survival analysis on the metastatic HMF cohort and found that both male and female patients had improved overall survival if their A/I ratio was above median (Figure 2d). We performed the same analysis on the individual cancer types, and found that 2/11 cancer types (BLCA and COAD) showed significantly improved prognosis with an A/I ratio above median, while no cancer types showed the opposite (Figure S6). When we performed a meta-analysis on male and female patients, respectively, we again found that an A/I ratio above median associated with improved outcome in both male and female patients (HR females 0.68, HR males 0.69, P < 0.05, Figure 2e-f).”

- “To investigate if the differences in survival were solely based on gender, we performed two cox proportional hazard models, one analysing survival relative to gender, and one analysing survival relative to gender and the A/I ratio. We then compared the performance of the models using a likelihood ratio test. Based on this analysis, we found that the model including the A/I ratio term significantly out-performed the simpler model including only gender (P = 4.45 * 10-9 237 )” -> I’m reading this as ignoring cancer type. Please remove, or at a minimum adjust for cancer type.

This analysis was actually adjusted for cancer types, unfortunately this was not clearly written, this has been corrected now. 

“To investigate if the differences in survival were solely based on gender, we performed two cox proportional hazard models, one analysing survival relative to gender, and one analysing survival relative to gender and the A/I ratio. Both models had age, stage and cancer type as covariates. ”

- Figure S7: Same as above: please remove or stratify by cancer type. (I believe the Mariathasan dataset is all bladder cancer; if so, then its results can be kept.)

The Mariathasan data is all bladder cancer. We have stratified the HMF results by cancertype, and inserted these as figure 7d-f. We have already included many large supplementary figures with km plots for both the TCGA and the HMF and their many different cancer types. Here, it is logical to include S7A+B in order to also illustrate the gender difference on a pan-cancer level. 

“To determine if the previously observed gender difference in cancer prognosis also affects survival within the two cohorts of CPI treated patients, we performed a survival analysis on gender. For both cohorts we found no significant difference in survival of male and and female patients treated by CPI (Mariathasan: P = 0.18, Figure S7c, HMF BLCA: P = 0.081, Figure S7d, HMF LUNG: P = 0.17, Figure S7e, HMF SKCM: P = 0.72, Figure S7f), indicating that drug-induced activation of the adaptive immune response may out-weigh any gender-specific differences in the immune response.”

- Line 280: “For both cohorts we found no significant difference in 281 survival of male and and female patients treated by CPI” -> the HMF results should be stratified by cancer type.

We have stratified the HMF results by cancertype, and inserted these as figure 7d-f.

“For both cohorts we found no significant difference in survival of male and and female patients treated by CPI (Mariathasan: P = 0.18, Figure S7c, HMF BLCA: P = 0.081, Figure S7d, HMF LUNG: P = 0.17, Figure S7e, HMF SKCM: P = 0.72, Figure S7f), indicating that drug-induced activation of the adaptive immune response may out-weigh any gender-specific differences in the immune response.”

- Figure 3b,d,f: please remove or stratify by cancer type.

Figure 3 is now re-labeled Figure 4. Figure 4b and 4d have been stratified by cancer type to allow for investigation of individual cancer types. Figure 4f is a summary and is kept as is. 

- “This supports our findings that metastatic melanoma with a poor response to CPI have a lower A/I ratio, indicating that they have a relatively high expression of the innate immune system.” -> This is a rather indirect inference. You should either remove this statement or back it up with an analysis of the innate immune score in the relevant samples.

Thank you for this comment, we agree that we could show this in a more direct manner, and we have therefore added a new figure to show this more clearly, Figure 5, which we have inserted below for your convenience. We have added the following text to the Result section:

“To further elucidate the relationship between expression of the adaptive and innate immune system in the tumour microenvironment of the CPI treated patients with progressive disease, we investigated the adaptive vs. the innate expression scores directly. Here, we observed that for both the HMF cohort and the Mariathasan bladder cancer, a very large proportion of patients with progressive disease showed higher innate relative to adaptive scores (BLCA: 96%, LUNG: 76 %, SKCM: 83%, Figure 5. Mariathasan BLCA: 79%, Figure S11). This supports the use of a ratio as a proper tool to identify patients with relatively increased adaptive-to-innate expression, and potentially improved therapy response” 

and to the Discussion:

“This supports our findings that 83% of patients with metastatic melanoma with a poor response to CPI have a lower A/I ratio, indicating that they have a relatively high expression of the innate immune system (Figure 5).” 

- It looks like your analyses found that innate abundance wasn’t informative when considered alongside adaptive abundance. At a minimum, please discuss the implications of this in the Discussion. Should we just be looking at adaptive abundance alone, or does your work provide additional reasons to look at A/I ratio?

It is our opinion that our work very convincingly shows that the ratio is better to utilize than the abundance alone. Considering innate and adaptive scores individually, there is no question that most information is found in the adaptive component. However, the ratio shows a more significant association, and on the new figure 5, we demonstrate how the higher innate abundance is associated with poor response to immunotherapy. Furthermore a ratio has the benefit that it is by definition normalised within each patient, thus we get a patient specific measurement that remains robust across patients and across cancer types.

---

## [Decision Letter · Decision Letter 2]

8 Jan 2023

PONE-D-22-08886R2The ratio of adaptive to innate immune cells differs between genders and associates with improved prognosis and response to immunotherapy

PLOS ONE

Dear Dr. Ahrenfeldt,

Thank you for submitting your manuscript to PLOS ONE. After careful consideration, we feel that it has merit but does not fully meet PLOS ONE’s publication criteria as it currently stands. Therefore, we invite you to submit a revised version of the manuscript that addresses the points raised during the review process.

We look forward to receiving your revised manuscript.

Kind regards,

Albert Rübben, Ass. Prof., M.D., Ph.D.

Academic Editor

PLOS ONE

Journal Requirements:

Additional Editor Comments:

I would like to congratulate the authors for their important contribution to cancer immunology and bioinformatic analysis. The last round of reviews have resulted in only minor requests which can be addressed by minor corrections and changes in wording.

The bottom line is the request to emphasize the limitations of the study with regard to pan-cancer analysis and to the need of future validation of the results.

I am looking forward to receiving the final version of the manuscript.

Yours sincerely

Albert Rübben

Reviewers' comments:

Reviewer's Responses to Questions

**Comments to the Author**

1. If the authors have adequately addressed your comments raised in a previous round of review and you feel that this manuscript is now acceptable for publication, you may indicate that here to bypass the “Comments to the Author” section, enter your conflict of interest statement in the “Confidential to Editor” section, and submit your "Accept" recommendation.

Reviewer #2: All comments have been addressed

Reviewer #3: All comments have been addressed

2. Is the manuscript technically sound, and do the data support the conclusions?

Reviewer #2: Yes

Reviewer #3: Yes

3. Has the statistical analysis been performed appropriately and rigorously? 

Reviewer #2: Yes

Reviewer #3: Yes

4. Have the authors made all data underlying the findings in their manuscript fully available?

Reviewer #2: Yes

Reviewer #3: Yes

5. Is the manuscript presented in an intelligible fashion and written in standard English?

Reviewer #2: Yes

Reviewer #3: Yes

6. Review Comments to the Author

Reviewer #2: This manuscript presents a carfully performed pan-cancer analysis, showing that the the ratio of adaptive to innate immune cells differs between females and males and associates with improved prognosis and response to immunotherapy. Using RNAseq data, an adaptive-to-innate immune ratio (A/I ratio) was defined and it was found with high significance that primary tumour samples from TCGA showed improved progression free survival in patients with an A/I ratio above median.

This is a quite interesting observation in these data sets, as the present standards for stratifying metastatic patients for immunotherapy (for e.g. via PDL1 expression or TMB measurement) is far from perfect. Thus, the proposede A/I ratio may be an additional suitable tool to identify patients which are likely to benefit from immunotherapy using checkpoint inhibitors. The gender specific aspect is also interesting and should be more carefully adressed by drug developers in future studies.

I think the manuscript is suitable for PLOS1, if the authors emphasise again that the observation is so far only based on pan-cancer data and needs further indepedant experimental confirmation. But as an introduction to this field of research (adaptive-to-innate immune ratio), it is certainly very exciting for others researchers as well.

Reviewer #3: (No Response)

7. PLOS authors have the option to publish the peer review history of their article (what does this mean?). If published, this will include your full peer review and any attached files.

Reviewer #2: No

Reviewer #3: No

---

## [Author Response · Author response to Decision Letter 2]

13 Jan 2023

Dear reviewers,

We strongly appreciate the time spent reviewing our manuscript. We have found your comments very useful, and we hope you will find that we have addressed them adequately. In our view, your comments have helped improve our manuscript, which we are very grateful for. 

Below you will find a point-by-point reply to each comment.

Reviewer 2

This manuscript presents a carefully performed pan-cancer analysis, showing that the the ratio of adaptive to innate immune cells differs between females and males and associates with improved prognosis and response to immunotherapy. Using RNAseq data, an adaptive-to-innate immune ratio (A/I ratio) was defined and it was found with high significance that primary tumour samples from TCGA showed improved progression free survival in patients with an A/I ratio above median.

This is a quite interesting observation in these data sets, as the present standards for stratifying metastatic patients for immunotherapy (for e.g. via PDL1 expression or TMB measurement) is far from perfect. Thus, the proposed A/I ratio may be an additional suitable tool to identify patients which are likely to benefit from immunotherapy using checkpoint inhibitors. The gender specific aspect is also interesting and should be more carefully addressed in future studies.

I think the manuscript is suitable for PLOS1, if the authors emphasise again that the observation is so far only based on pan-cancer data and needs further independent experimental confirmation. But as an introduction to this field of research (adaptive-to-innate immune ratio), it is certainly very exciting for others researchers as well.

Thank you very much for your comments and for reviewing our manuscript. We have added an extra paragraph to the discussion, where we emphasize the need for experimental validation.

“As the results in this study are based on publicly available pan-cancer data from heterogeneous cohorts, independent experimental validation would be necessary to further validate using a ratio of adaptive to innate immune cells for patient stratification.”

Reviewer 3

The manuscript has already been through many revisions, and thus I think it is in good shape. It reads well and the concepts, ideas and evidence are clear. The authors offer a concise introduction to immunotherapy and provide a solid base for their rationale. Furthermore, I think that enough evidence is provided in favour of the hypothesis that the type of immune cell infiltration is important for patient stratification. While more statistical power is necessary to provide final evidence, we applaud the authors’ efforts to mine currently available data. Advances in machine learning and statistical extrapolation, combined with the increasing number of patient data could take this concept into a fully-fledged clinical tool in the not-too- distant future.

Before publication, we would kindly ask the author to address the following:

Could you please mention in the introduction the cell types that Adaptive immune cell types (CD8 T-cells, B-cells, CD45, Cytotoxic cells, T-cells, Th1-cells and T-regulatory cells) and innate immune cell types (Dendritic cells, Macrophages, Mast cells, Neutrophils, Natural killer cells and Natural killer CD56dim cells).  

Thank you for this very sensible suggestion, we humbly apologise for not already having included such a section. We agree that it improves understanding to have these mentioned in the introduction. We have therefore added the following paragraphs to the introduction, and included two new references (11 and 12).

“The immune system can roughly be divided into two major branches, the innate and the adaptive. The innate immune system is our first line of defence, but it is non-specific and its primary role is to initiate inflammation when recognizing foreign pathogens, and to use phagocytosis to engulf foreign molecules and cells, and then present antigens from these to the cells of the adaptive immune system that can activate a specific immune response[11]. The adaptive immune system contains cells that undergo recombination to create unique receptors which bind to foreign peptides or peptides not usually presented by normal, healthy cells[12].”

“For this study, Dendritic cells, Macrophages, Mast cells, Neutrophils, Natural killer cells and Natural killer CD56dim cells were all analysed as part of the innate immune system. Likewise CD8 T-cells, B-cells, CD45, Cytotoxic cells, T-cells, Th1-cells and T-regulatory cells were all analysed as part of the adaptive immune system [11], [12].”

Could you please fix the following typos and grammar:

Ln 20: ... “are” far from perfect

This has been corrected.

Ln 49: ... “plays” an .. .

This has been corrected.

Ln 68: A great amount of research is “being” performed ...

This has been corrected.

Ln 82: please lowercase “immune system”

This has been corrected.

Ln 110: needs a new line to separate paragraphs

New lines have been added both above and below, to separate the three separate paragraphs.

Ln 135: Please isolate in a new line the equation:

scaled_celltypen = (celltypen - celltypemin ) / (celltypemax - celltypemin) (eq1)

This has been correctly formatted as an equation.

Ln 196- why are the abbreviated cancers not in alphabetical order? Is there a significance to the current order? It would be easier to look them up in Table 1 if they were.

We have changed the order to be alphabetical. They were ordered from lowest to highest p-value, the same order in which they appeared on the figure, but we agree that it is easier to look them up, when they are in alphabetical order.

Line 326- I would suggest softening the claim by using the word “useful” or “efficient” instead of “proper”.

Thank you for this suggestion, we changed the word to “efficient”, as we agree that this better expresses the meaning of the sentence.

Line 360- Same here. I would suggest to use the words “more accurately” instead of “properly”.

Thank you for this suggestion, we have changed the words to “more accurately”, as we again agree that this better expresses the meaning of the sentence.

Line 381 - Did you mean that “ , whereas here a low frequency ...”

No, that sentence was referring to the findings in the paper, and to make this clear we have changed the sentence to

“A study reports that the baseline circulating myeloid-derived suppressor cells (MDSC) correlate with outcome of ipilimumab treatment[45], they find that a low frequency correlates to improved outcome.”

Figure 3- some cancers are missing label. Could you create a legend that goes next to it for those missing a label?

We have only labelled the significantly different cancer types. A legend has been added, to show the colour of all cancers, as we agree that this improves the understanding of the figure.

Figure general- the margins of the figures are too small, and some of the figure text overlaps with text generated by the automatic review pdf print.

Thank you for noticing this! We have made all the margins wider, to avoid any overlaps.

---

## [Decision Letter · Decision Letter 3]

23 Jan 2023

The ratio of adaptive to innate immune cells differs between genders and associates with improved prognosis and response to immunotherapy

PONE-D-22-08886R3

Dear Dr. Ahrenfeldt,

We’re pleased to inform you that your manuscript has been judged scientifically suitable for publication and will be formally accepted for publication once it meets all outstanding technical requirements.

Kind regards,

Albert Rübben, Ass. Prof., M.D., Ph.D.

Academic Editor

PLOS ONE

Additional Editor Comments (optional):

The authors have presented a highly interesting study with significant implications for the fields of cancer immunology and cancer bioinformatics.

Reviewers' comments:

Reviewer's Responses to Questions

**Comments to the Author**

1. If the authors have adequately addressed your comments raised in a previous round of review and you feel that this manuscript is now acceptable for publication, you may indicate that here to bypass the “Comments to the Author” section, enter your conflict of interest statement in the “Confidential to Editor” section, and submit your "Accept" recommendation.

Reviewer #2: All comments have been addressed

Reviewer #3: All comments have been addressed

2. Is the manuscript technically sound, and do the data support the conclusions?

Reviewer #2: Yes

Reviewer #3: Yes

3. Has the statistical analysis been performed appropriately and rigorously? 

Reviewer #2: Yes

Reviewer #3: Yes

4. Have the authors made all data underlying the findings in their manuscript fully available?

Reviewer #2: Yes

Reviewer #3: Yes

5. Is the manuscript presented in an intelligible fashion and written in standard English?

Reviewer #2: Yes

Reviewer #3: Yes

6. Review Comments to the Author

Reviewer #2: The authors have taken my suggestions satisfactorily and the paper is now acceptable to PLOS1. xxxxxxxxxxxxxxxxxxxxxxxxxxxxxxxxxxxxxxxxxxxxxxxxxxxxxxxxxxxxxxxxxxxxxxxxxxxxxxxxxxxxxxxxxxxxxxxxxxxxxxxxxxxxxxxxxxxxxxxxxxxxxxxxxxxxxxxxxxxxxxxxxxxxxxxxxxxxxxxx

Reviewer #3: (No Response)

7. PLOS authors have the option to publish the peer review history of their article (what does this mean?). If published, this will include your full peer review and any attached files.

Reviewer #2: No

Reviewer #3: No

---

## [Editor Report · Acceptance letter]

26 Jan 2023

PONE-D-22-08886R3 

The ratio of adaptive to innate immune cells differs between genders and associates with improved prognosis and response to immunotherapy 

Dear Dr. Ahrenfeldt:

I'm pleased to inform you that your manuscript has been deemed suitable for publication in PLOS ONE. Congratulations! Your manuscript is now with our production department. 

Kind regards, 

on behalf of

Albert Rübben 

Academic Editor

PLOS ONE